# Inducible mechanisms of disease tolerance provide an alternative strategy of acquired immunity to malaria

**Wiebke Nahrendorf[1]\*, Alasdair Ivens[1,2], Philip J Spence[1,2]\***

[1]Institute of Immunology and Infection Research, University of Edinburgh, Edinburgh, United Kingdom; [2]Centre for Immunity, Infection and Evolution, University of Edinburgh, Edinburgh, United Kingdom

**Abstract** Immunity to malaria is often considered slow to develop but this only applies to defense mechanisms that function to eliminate parasites (resistance). In contrast, immunity to severe disease can be acquired quickly and without the need for improved pathogen control (tolerance). Using *Plasmodium chabaudi*, we show that a single malaria episode is sufficient to induce host adaptations that can minimise inflammation, prevent tissue damage and avert endothelium activation, a hallmark of severe disease. Importantly, monocytes are functionally reprogrammed to prevent their differentiation into inflammatory macrophages and instead promote mechanisms of stress tolerance to protect their niche. This alternative fate is not underpinned by epigenetic reprogramming of bone marrow progenitors but appears to be imprinted within the remodelled spleen. Crucially, all of these adaptations operate independently of pathogen load and limit the damage caused by malaria parasites in subsequent infections. Acquired immunity to malaria therefore prioritises host fitness over pathogen clearance.

**\*For correspondence:**
Wiebke.Nahrendorf@ed.ac.uk
(WN);
Philip.Spence@ed.ac.uk (PJS)

**Competing interests:** The authors declare that no competing interests exist.

## Introduction

Mechanisms of host resistance can eliminate pathogens, but it is disease tolerance that functions to preserve life. Tolerance mechanisms of host defense do not have a direct impact on pathogen load and instead act to minimise tissue damage caused by the pathogen and the immune response targeting it. They also function to protect vital homeostatic processes, such as energy metabolism, under conditions of infection-induced stress (*Martins et al., 2019*). We have taken tremendous steps to understand acquired resistance mechanisms that can eliminate pathogens and provide sterile immunity. On the other hand, it is unclear whether mechanisms of disease tolerance can persist after pathogen clearance to provide an alternative strategy of acquired immunity. And it is these mechanisms that are likely to be at the forefront of host defense when sterile immunity cannot be generated.

We propose that immunity to severe life-threatening malaria is underpinned by acquired mechanisms of disease tolerance. The majority of malaria-induced deaths occur in children under the age of 5 infected with *Plasmodium falciparum* (*Weiss et al., 2019*). A landmark prospective study in Tanzania followed 882 children from birth and showed that the risk of developing severe malaria is highest in the first few infections of life, and very few children (<1.8%) have more than one severe episode (*Gonçalves et al., 2014*). These data therefore support the longstanding view that immunity against severe malaria is acquired rapidly – often before 12 months of age (*Gupta et al., 1999*; *Marsh and Snow, 1999*). Crucially, this study further showed that children who survive severe malaria are frequently reinfected and experience episodes of febrile malaria with similar or even higher parasite densities (*Gonçalves et al., 2014*). Immunity to severe forms of malaria is therefore

**eLife digest** Malaria is a parasitic infection spread by mosquitoes that causes hundreds of millions of cases each year. People are most likely to die from malaria the first time they are infected – usually when they are young children. Among those who survive, however, few will develop severe symptoms again, even though they are often reinfected with as many (or even more) parasites. This indicates that people do not get better at eliminating the parasite. Instead, protection from severe malaria is a form of tolerance - the body learns to limit the damage the infection causes. But exactly which mechanisms have to be engaged to tolerate malaria is unclear.

One way to achieve tolerance may be to switch off damaging inflammation. Nahrendorf et al. explored this possibility by comparing the immune response of mice to their first and second infection with malaria parasites. During the first infection of life, immune cells release harmful inflammatory molecules that activate the lining of blood vessels, causing tissue damage and severe symptoms. During the second infection, these immune cells shut down inflammation and instead actively promote tissue health to reduce damage and improve outcome. This change in the immune response occurs despite the fact that the number of parasites is the same in both infections.

Nahrendorf et al. also found that the mouse's immune cells 'remembered' to tolerate subsequent infections, even after treatment with a drug that kills all malaria parasites. This was possible because malaria permanently altered the spleen, which reprogrammed the response of the immune cells. A single infection is therefore enough to induce long-lived mechanisms of tolerance that can prevent life-threatening disease.

These findings have the potential to change the understanding of immunity to malaria, which currently emphasises the importance of killing parasites. New ways to treat and vaccinate people - and to protect young children from severe malaria - may arise by treating tolerance as an equally important form of host defense.

not due to improved parasite elimination (resistance) but instead underpinned by the improved ability of the host to limit the pathological consequences of infection (tolerance).

There is a growing body of evidence that shows mechanisms of disease tolerance are required to survive a first malaria episode: for example, acute infection causes hypoglycaemia and so the ability to maintain blood glucose levels within dynamic range – in crosstalk with iron metabolism (*Weis et al., 2017*) – can determine life or death (*Cumnock et al., 2018*). Furthermore, the induction of heme oxygenase-1 by nitric oxide (*Jeney et al., 2014*) leads to the detoxification of free heme, which is released from ruptured red cells (*Seixas et al., 2009*) – this protects renal proximal tubule epithelial cells and prevents acute kidney injury (*Ramos et al., 2019*). Nonetheless, whilst these metabolic adaptations are doubtless essential for the survival of naive hosts there is no evidence as yet that they can be recalled and applied more effectively in subsequent infections to provide clinical immunity.

Instead, the most effective way to induce persisting mechanisms of disease tolerance may be through host control of inflammation (*Medzhitov et al., 2012*), which can limit collateral tissue damage and avert fatal metabolic perturbations. In malaria, the ability to control systemic inflammation may also minimise the detrimental effects of parasite sequestration by reducing activation of the endothelium and restricting available binding sites in the microvasculature (*Schofield and Grau, 2005*). Field data support the idea that dampening the inflammatory response to *P. falciparum* affords protection; lower levels of circulating pro-inflammatory cytokines are found in children in Malawi who survive severe malaria compared to those who succumb (*Mandala et al., 2017*). And critically, inflammation can be reduced in the absence of improved parasite control; Ghanaian children in a high transmission area have higher parasite densities but less systemic inflammation and fewer febrile episodes than children in a lower transmission setting (*Ademolue et al., 2017*).

The blood cycle in malaria unleashes a plethora of parasite-derived as well as host tissue damage-associated signals, all of which can be sensed by innate immune cells. Additionally, pronounced changes in host physiology (such as hypoxia and acidaemia) are hallmarks of severe disease (*von Seidlein et al., 2012*). Together these diverse signals trigger monocytes and macrophages to produce pro-inflammatory molecules, many of which have been associated with a poor prognosis

including the prototypical myeloid-derived cytokines TNF and IL-6 (*Mandala et al., 2017*). Macrophages can even directly cause severe anaemia through extensive bystander phagocytosis of healthy uninfected red cells in the spleen and impairment of erythropoiesis in the bone marrow (*Jakeman et al., 1999*; *Pathak and Ghosh, 2016*). Given these key roles in pathogenesis, it may be possible to quickly reduce the risk of severe malaria by modifying the myeloid response to blood-stage infection. Importantly, an emerging body of literature shows that the response of myeloid cells is not hardwired but can be reprogrammed by pathogens and their products. This was first demonstrated by stimulating bone marrow-derived macrophages with LPS in vitro, which reduces the production of pro-inflammatory cytokines and increases the release of antimicrobial effector molecules upon re-stimulation (*Foster et al., 2007*). The ability of monocytes and macrophages to adapt their response to repeated pathogen encounters through cell intrinsic modifications is termed innate memory (*Netea et al., 2016*). And although myeloid cells are usually short-lived and terminally differentiated, memory can nevertheless be imprinted through the epigenetic reprogramming of either long-lived tissue-resident macrophages (*Wendeln et al., 2018*) or progenitor cells in the bone marrow (*Kaufmann et al., 2018*).

Innate memory has been shown to operate independently of pathogen load (*Dominguez-Andres and Netea, 2019*; *Seeley and Ghosh, 2017*) and human monocytes produce less pro-inflammatory cytokines when stimulated after an episode of febrile malaria as compared to before (*Portugal et al., 2014*), suggesting that they are intrinsically modified by infection. We therefore hypothesised that innate memory – leading to the functional specialisation of monocytes and macrophages to limit inflammation and associated pathology – offers the most compelling explanation for how immunity to severe malaria can be acquired so quickly and without the need for enhanced parasite control.

## Results

To investigate acquired mechanisms of disease tolerance and the role of innate memory in vivo, we needed to examine monocyte progenitors in the bone marrow and long-lived tissue-resident macrophages in the spleen. And since these tissues are not readily accessible in human malaria, we needed a model that would recapitulate at least some key features of human infection. Given that a meta-analysis of malariotherapy data shows naive hosts quickly adapt to tolerate chronic parasitaemia (for example, by increasing their pyrogenic threshold [*Gatton and Cheng, 2002*]), we chose a rodent malaria parasite (*Plasmodium chabaudi*) that establishes chronic recrudescing infections in laboratory mice. Importantly, experimental infections were initiated with sporozoites, since we have previously shown that mosquito transmission resets expression of the large sub-telomeric multi-gene families that control parasite virulence (*Spence et al., 2013*). And furthermore, we used two parasite genotypes to try and uncouple the relative contribution of parasite-derived versus damage-associated signals in promoting mechanisms of tolerance. *P. chabaudi* AS causes a mild infection, characterised by a low pathogen load and few clinical symptoms (*Figure 1A–C*). In contrast, *P. chabaudi* AJ (which has more than 140,000 SNPs cf. *P. chabaudi* AS [*Otto et al., 2014*]) leads to acute hyperparasitaemia and severe anaemia (*Figure 1B–C*), accompanied by hypothermia and prostration (*Figure 1— figure supplement 1A*). AJ shares many key features with AS such as synchrony, chronicity and a persisting low-grade anaemia (*Figure 1B–C*, *Figure 1—figure supplement 1B–C*), and yet whilst AS sequesters in key immune sites such as bone marrow and spleen (*Brugat et al., 2014*) we find no evidence that AJ sequesters in host tissues (*Figure 1—figure supplement 1D*).

### Malaria triggers emergency myelopoiesis and obliterates tissue-resident macrophages

To ask whether malaria can functionally reprogramme myeloid cells, we must first understand their response to acute infection in a naive host; we started by mapping their dynamics in our severe model of disease. We found that the bone marrow quickly prioritises myelopoiesis by increasing the number of granulocyte monocyte progenitors (GMP) (*Figure 1D*, see *Supplementary file 1* for gating strategies). Consequently, an increased number of inflammatory monocytes and neutrophils are released into circulation and recruited into their target organ – the spleen (*Figure 1D* and *Figure 1— figure supplement 2A*). Furthermore, megakaryocyte erythroid progenitors (MEP) appear de novo in the spleen (*Figure 1E*); this extramedullary mechanism of erythropoiesis likely represents a

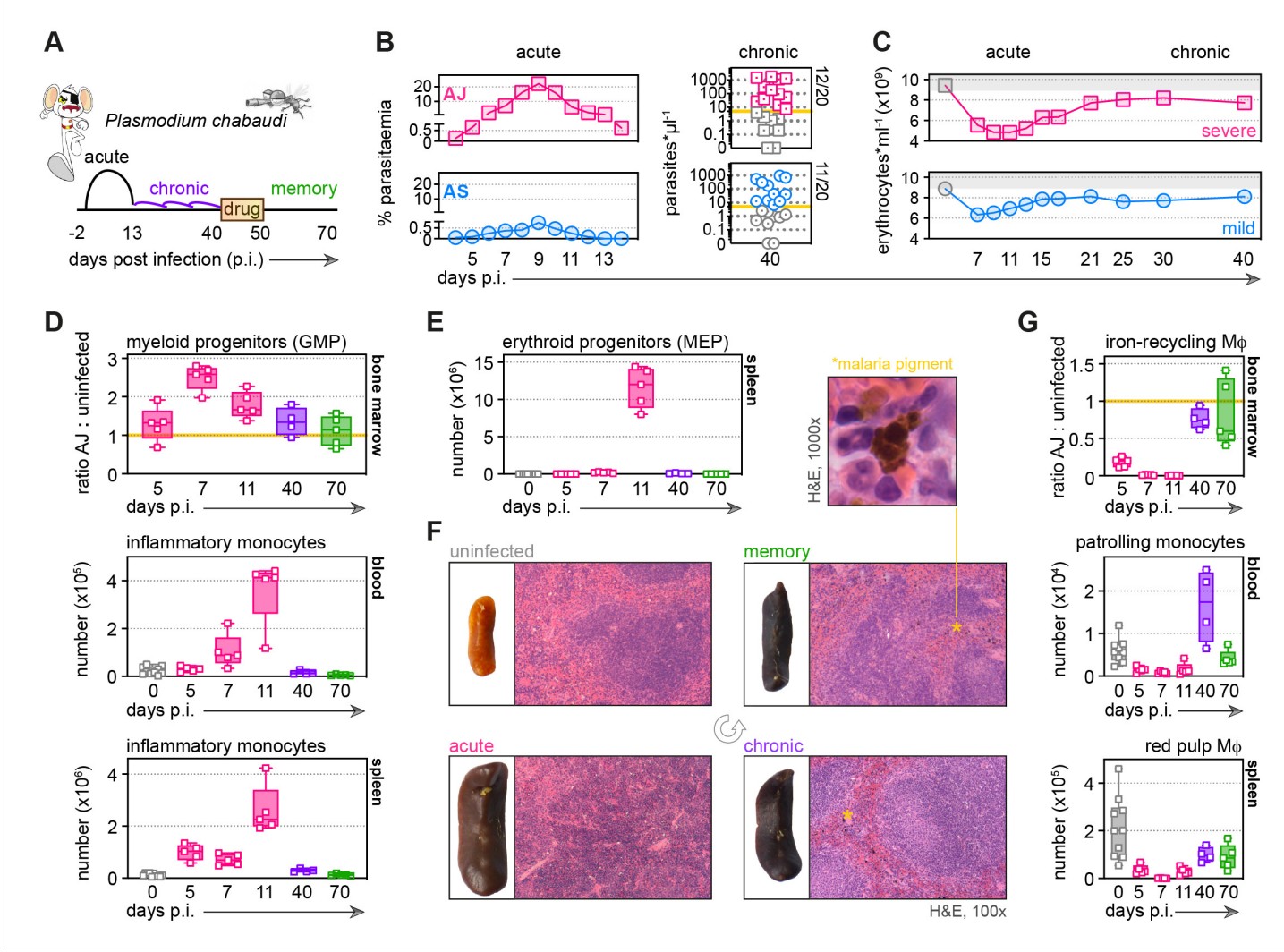

**Figure 1.** Malaria triggers emergency myelopoiesis and obliterates tissue-resident macrophages. (**A**) C57Bl/6 mice were infected with *Plasmodium chabaudi* AJ or AS sporozoites; the blood-stage of infection started 2 days later after the release of merozoites from the liver. Mice were chronically infected for 40 days, at which point we administered the antimalarial drug chloroquine. Memory responses were assessed 30 days thereafter. Note that we exclusively use days post infection (p.i.) to refer to the blood-stage of malaria. (**B**) Acute parasitaemia was monitored daily using Giemsa stained thin blood films and chronic infection was verified 40 days p.i. by qPCR (n = 20 per group). Symbols below the limit of detection (5 parasites*$\mu$l$^{-1}$) are coloured grey and these mice were excluded from the study. (**C**) The mean number of erythrocytes*ml$^{-1}$ is shown before (grey symbols) and during infection (n = 10 for AJ and n = 14 for AS). Severe anaemia is defined as >50% loss of red cells. (**D and E**) Inflammatory monocytes and progenitor cells (granulocyte monocyte progenitors [GMP] and megakaryocyte erythroid progenitors [MEP]) from uninfected mice (0 days p.i.), AJ-infected mice (5, 7, 11, and 40 days p.i.) and once-infected mice (memory, 70 days p.i.) were analysed by flow cytometry (n = 4–5 per time point, box-plots show median and IQR). Uninfected age-matched controls were analysed at each time point and pooled for graphing (n = 10); absolute counts are shown for blood and spleen. In (**D**), GMP are shown as a ratio of infected:uninfected at each time point because bone marrow cellularity increases with age. (**F**) Paraffin-embedded spleen sections were H&E stained (11 days p.i. for acute AJ infection) – examples of malaria pigment in chronically infected and once-infected mice are marked with an asterisk. (**G**) Tissue-resident macrophages (M$\Phi$) and patrolling monocytes from uninfected mice, AJ-infected mice, and once-infected mice were analysed by flow cytometry (n = 4–5 per time point, box-plots show median and IQR). Absolute counts (blood and spleen) and cell ratios (bone marrow) are shown exactly as described for (**D and E**). See *Supplementary file 1* for all antibody panels and gating strategies. The online version of this article includes the following figure supplement(s) for figure 1:

**Figure supplement 1.** *P. chabaudi* AJ causes severe disease without sequestering in host tissues.
**Figure supplement 2.** Malaria causes major disturbances in tissue structure and integrity.

division of labour in an attempt to compensate for the loss of erythroid progenitors in the bone marrow (*Pathak and Ghosh, 2016*). We also observed major histological changes in tissue structure and integrity with reduced cellularity in the bone marrow contrasting starkly with marked splenomegaly, which was accompanied by a complete loss of organisation between red and white pulp (*Figure 1F* and *Figure 1—figure supplement 2B–C*).

Remarkably, we found that long-lived prenatally seeded tissue-resident macrophages in bone marrow and spleen (*Hashimoto et al., 2013*) rapidly disappear during acute infection (*Figure 1G*). Since red pulp macrophages are the only cells that can store and recycle iron in the spleen their disappearance thus means that ferric iron, which can be revealed histologically with Prussian Blue staining, is completely absent at the peak of infection (*Figure 1—figure supplement 2D–E*). We could further demonstrate that patrolling monocytes (*Carlin et al., 2013*), often regarded as the tissue-resident macrophages of the vasculature (*Mildner et al., 2017*), also disappear early in infection (*Figure 1G*). These findings therefore place inflammatory monocytes at the centre of the acute phase response, since they now provide the only route through which to phagocytose and clear infected red cells.

## Monocytes differentiate into inflammatory macrophages in naive hosts

We therefore carefully characterised the fate and function of inflammatory monocytes in the spleen by RNA sequencing (*Figure 2A*) and used clueGO to reveal the complexity of their response to a first encounter with malaria parasites (*Bindea et al., 2009*; *Mlecnik et al., 2014*). ClueGO assigns significant gene ontology (GO) terms based on differential gene expression and groups them into functional networks by relatedness. When we merged all linked nodes into supergroups (see Materials and methods) we found that more than one third of all GO terms were related to host defence (*Figure 2B and C*). Furthermore, clueGO identified interferon signaling as an upstream regulator of monocyte fate (*Figure 2B*); in agreement, interferon-inducible guanylate binding proteins were highly upregulated (*Figure 2—figure supplement 1A*).

We next used core lineage signatures to predict the likely outcome of monocyte differentiation in the spleen, where they can be instructed to become either inflammatory macrophages or monocyte-derived dendritic cells (*Menezes et al., 2016*). This revealed that monocytes initiate a transcriptional programme that is typical of terminally differentiated inflammatory macrophages (*Figure 2D*). They upregulate their ability to sense parasite- and host-derived danger signals by increasing transcription of diverse pattern recognition receptors (*Figure 2—figure supplement 1B*), and upregulate expression of the hallmark cytokines and chemokines associated with a type I inflammatory response (*Figure 2E*). Furthermore, they enhance their capacity to engage T cell immunity by upregulating all major components of the antigen processing and presentation machinery (for class I and class II MHC), and attempt to fine-tune T cell activation by increasing their expression of co-stimulatory molecules and checkpoint inhibitors (*Figure 2F–G*). Notably, a clear Warburg effect – the metabolic switch from oxidative phosphorylation to glycolysis described when monocytes are stimulated with LPS in vitro (*Cheng et al., 2014*) – was not observed in vivo in response to malaria. Instead, the key enzymes involved in both metabolic pathways were transcriptionally induced (*Figure 2—figure supplement 1C–D*).

We next looked at all of these parameters of monocyte and macrophage biology in our mild model of malaria. Surprisingly, despite substantial differences in parasite density and patterns of sequestration (*Figure 1B* and *Figure 1—figure supplement 1D*) the response of myeloid and progenitor cells in bone marrow, blood, and spleen was remarkably similar between AS and AJ (*Figure 2—figure supplement 2A–C*). Moreover, the fate and function of spleen monocytes was essentially identical – a direct pairwise comparison identified only a single differentially expressed gene (*Kelch34*) between the two models. In turn, clueGO analysis revealed that the four largest supergroups in monocytes isolated from AJ-infected mice (*Figure 2B–C*) also dominated the response to AS (*Figure 2—figure supplement 2D–E*). It therefore appears that parasite genotype, pathogen load, and the sequestration of infected red cells in immune tissues does not fundamentally alter the myeloid response to acute infection. Instead, the haematopoietic switch that promotes myelopoiesis in the bone marrow (and relocates erythropoiesis to the spleen), the disappearance of tissue-resident macrophages and the differentiation of monocytes into inflammatory macrophages may all be part of an emergency response that is unavoidable in a naive host.

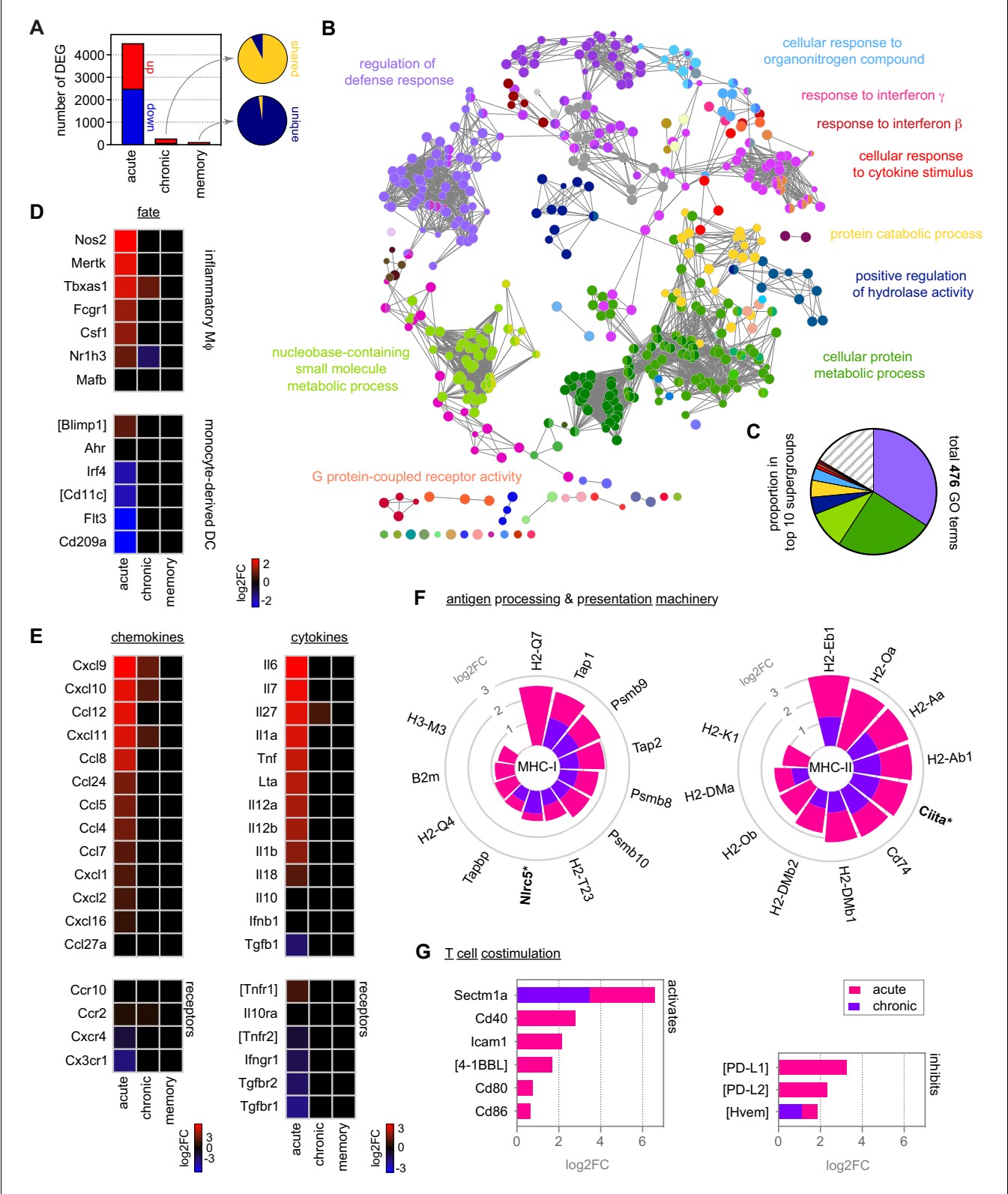

**Figure 2.** Monocytes differentiate into inflammatory macrophages in naive hosts but become quiescent in chronic infection. (**A**) RNA sequencing of spleen monocytes flow-sorted from AJ-infected mice (7 and 40 days p.i.for acute and chronic, respectively) and once-infected mice (memory, 70 days p.i.). At each time point, the number of differentially expressed genes (DEG, $p_{adj}$ <0.01, >1.5-fold change) was assessed relative to uninfected controls. Pies show the proportion of shared or unique DEG between chronic and acute infection (top) or memory and chronic infection (bottom). (**B and C**)

*Figure 2 continued on next page*

*Figure 2 continued*

ClueGO network of DEG in spleen monocytes during acute infection. Each node represents a significantly enriched gene ontology (GO) term and node size is determined by $p_{adj}$. Related GO terms that share >40% of genes are connected by a line and organised into functional groups (each given a unique colour). Supergroups are formed when GO terms are shared between more than one group. The names of the top 10 (super)groups (lowest $p_{adj}$) are displayed and (C) shows their proportion of total GO terms. (D–G) Log2FC of (D) signature genes used to predict monocyte differentiation into inflammatory macrophages (MΦ) or monocyte-derived dendritic cells (DC); (E) chemokines, cytokines and their receptors; (F) class I and class II antigen processing and presentation pathways (inc MHC transactivators*); and (G) T cell co-stimulation and inhibitory ligands. At each time point, log2FC is shown relative to uninfected controls. Square brackets indicate that common gene names were used. At each time point in (A–G), n = 5–6 for infected mice and n = 6–7 for uninfected controls.

The online version of this article includes the following figure supplement(s) for figure 2:

**Figure supplement 1.** Monocytes upregulate glycolysis and oxidative phosphorylation as they differentiate into inflammatory macrophages.

**Figure supplement 2.** Parasite genotype does not influence the emergency myeloid response to malaria.

**Figure supplement 3.** Quiescent monocytes can differentiate into inflammatory macrophages when removed from the spleen.

## Naive hosts adapt to tolerate chronic infection

Malaria parasites can persist for many months (or even years) in humans (*Felger et al., 2012*) and the fitness costs of maintaining emergency myelopoiesis over these time frames would be exceptionally high. We therefore asked how the host adapts to an ongoing infection that can not be cleared. In the chronic phase of *P. chabaudi*, the pathogen load can reach up to 1000 parasites per µl blood (*Figure 1B*) and insoluble malaria pigment accumulates throughout the red pulp of the spleen (*Figure 1F*). Despite this abundance of parasite-derived signals, the spleen stops extramedullary erythropoiesis and creates new structural demarkations between red and white pulp (*Figure 1E–F*). Furthermore, the bone marrow reduces the production of granulocyte monocyte progenitors, which in turn reduces the number of inflammatory monocytes and neutrophils trafficking into the blood and spleen (*Figure 1D* and *Figure 1—figure supplement 2B*). At the same time, resident macrophages begin to repopulate their tissue niches (*Figure 1G*) and ferric iron stores are re-established in the spleen (*Figure 1—figure supplement 2E*). These data provide compelling evidence that naive hosts quickly adapt to tolerate malaria parasites and return the myeloid compartment towards a healthy uninfected baseline.

In agreement, the transcriptome of monocytes during chronic infection is almost indistinguishable from uninfected controls (*Figure 2A*). Monocytes are no longer programmed to differentiate into inflammatory macrophages upon their arrival in the spleen (*Figure 2D*) and they silence transcriptional signatures of the acute phase response (*Figure 2E–G* and *Figure 2—figure supplement 1A–B*). Notably, we find no evidence that they engage mechanisms to suppress inflammation, such as regulatory cytokines (IL-10 and TGFβ) or inhibitors of T cell activation (PD-L1) (*Figure 2E and G*). And furthermore, we find no evidence that they have an alternative fate such as inflammatory hemophagocytes (*Figure 2—figure supplement 1E*), which have been implicated in chronic anaemia (*Akilesh et al., 2019*). Instead, monocytes simply adopt a state of quiescence during chronic infection despite persisting parasitaemia.

Importantly, we can show that quiescence is reversible – when monocytes are removed from the spleen of chronically infected mice and stimulated in vitro with LPS their inflammatory response is comparable to monocytes from uninfected mice (*Figure 2—figure supplement 3A–B*). In both cases, we clearly observe the Warburg effect and transcription of a plethora of inflammatory cytokines, chemokines and co-stimulatory molecules, together with the induction of *Nos2* – the best biomarker of an inflammatory macrophage fate (*Figure 2—figure supplement 3C–D*). Clearly, monocytes are not in a permanent refractory state during chronic infection; instead, their activation and differentiation in response to parasites and their pyrogenic products must somehow be silenced in the remodelled spleen. This likely represents just one mechanism through which the host minimises inflammation to resolve collateral tissue damage.

## Tolerance persists to protect host tissues during reinfection

We next asked whether tolerance could persist in the absence of live replicating parasites to provide long-lived protection. To answer this question, we developed a novel reinfection model that allowed us to exactly match parasite densities between first and second infection. To this end, mice were first infected with the avirulent parasite genotype AS to induce chronic recrudescing parasitaemia and

then drug-treated after 40 days of infection to clear circulating and sequestered parasites. One month later, mice were infected for a second time but now with the virulent genotype AJ (*Figure 3A*). In this model, parasite burden (*Figure 3B*) and the dynamics of red cell loss (*Figure 3C*) are both matched between first and second infection – this eliminates pathogen load as a confounding factor when analysing acquired mechanisms of disease tolerance.

In contrast to a first malaria episode, the bone marrow does not prioritise the production of myeloid cells upon reinfection (*Figure 3D*) and preserves its cellularity and structural integrity (*Figure 3—figure supplement 1A*). Consequently, the number of inflammatory monocytes and neutrophils released into circulation does not increase and nor does their accumulation in the spleen (*Figure 3E* and *Figure 3—figure supplement 1B*). Furthermore, the spleen maintains its boundaries between red and white pulp and does not promote extramedullary erythropoiesis (*Figure 3D* and *Figure 3—figure supplement 1C*). And whilst first infection obliterates tissue-resident macrophages these cells are resistant to malaria-induced ablation second time around (*Figure 3F*); this allows the host to retain ferric iron stores (*Figure 3—figure supplement 1D*). A single malaria episode is therefore sufficient to disarm emergency myelopoiesis in the bone marrow and protect terminally differentiated macrophages in the spleen. What's more, tissue architecture is preserved and key homeostatic processes are maintained in tolerised hosts.

## Monocytes minimise inflammation and stress in tolerised hosts

Nevertheless, we reasoned that the myeloid compartment cannot be entirely quiescent during reinfection since mice are able to control replication of the virulent parasite genotype AJ. We therefore

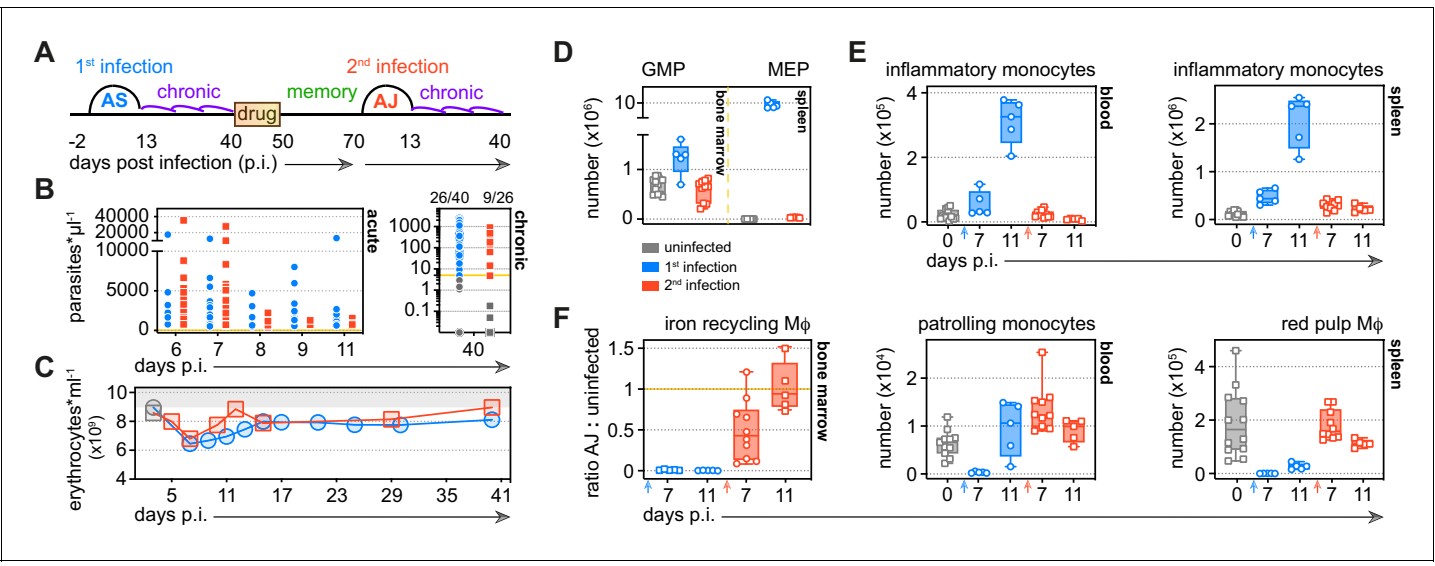

**Figure 3.** Tolerance persists to protect host tissues during reinfection. (A) Malaria reinfection model: C57Bl/6 mice were first infected with *P. chabaudi* AS sporozoites. Chronic infection was confirmed by qPCR at 40 days p.i. and drug-treated with the antimalarial drug chloroquine. Thirty days thereafter mice were infected for a second time with *P. chabaudi* AJ. (B) Circulating parasite density was calculated using percentage parasitaemia (limit of detection 0.01%) and red cell counts, and is presented as the number of parasites*μl$^{-1}$ throughout the acute phase of first and second infection. No statistically significant difference was detected at any timepoint (p$_{adj}$ <0.05, Mann-Whitney test corrected for multiple comparisons using Holm-Šidák method). Chronic infection was verified 40 days p.i. by qPCR (n = 40 in 1st infection and n = 26 in 2nd infection) and symbols below the limit of detection (5 parasites*μl$^{-1}$) are coloured grey; these mice were excluded from the study. (C) The mean number of erythrocytes*ml$^{-1}$ is shown before (grey symbols) and during first and second infection (n = 14 per group). (D–F) Inflammatory and patrolling monocytes, progenitors and tissue-resident macrophages (MΦ) from mice experiencing their first (n = 5 per time point) or second (n = 5–10 per time point) infection were analysed by flow cytometry (box-plots show median and IQR). Uninfected age-matched controls were analysed at each time point and pooled for graphing (n = 12); absolute counts are shown except for tissue-resident MΦ in the bone marrow. In this case, data are presented as a ratio of infected:uninfected at each time point. In (D), granulocyte monocyte progenitors (GMP) and megakaryocyte erythroid progenitors (MEP) are shown 11 days p.i. See *Supplementary file 1* for all antibody panels and gating strategies.

The online version of this article includes the following figure supplement(s) for figure 3:

**Figure supplement 1.** Tissue architecture is preserved and key homeostatic processes are maintained in tolerised hosts.

isolated spleen monocytes and examined their transcriptional programme in the acute phase of second infection. We identified more than 3000 differentially expressed genes and found that most of these were unique to reinfection (*Figure 4A*); remarkably, monocytes did not differentiate into inflammatory macrophages and all functions associated with this fate were silenced (*Figure 4B–C*, *Figure 4—figure supplement 1A–C*). Host control of inflammation extended beyond the boundaries of the spleen with circulating levels of CXCL10 and IFNɣ also attenuated at the peak of second infection (*Figure 4D*).

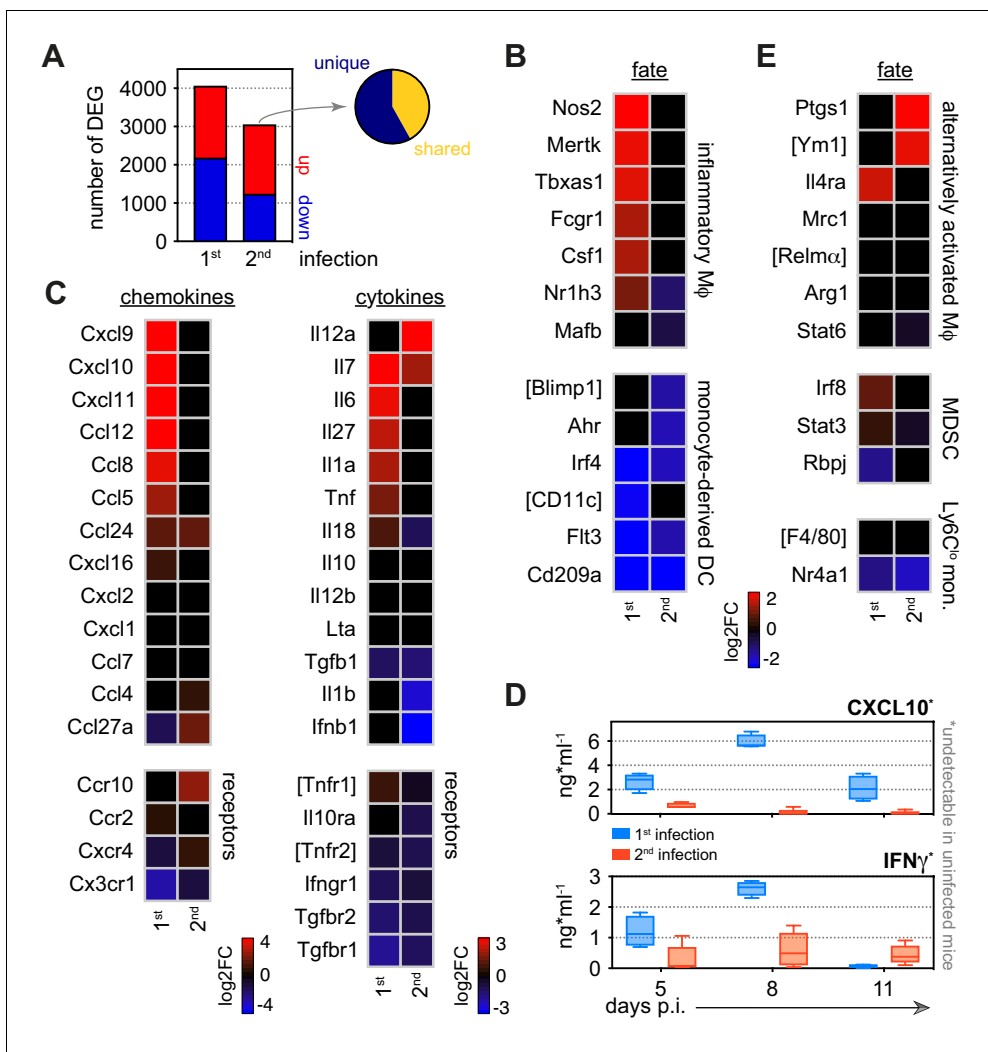

**Figure 4.** Monocytes minimise inflammation in tolerised hosts. (**A**) RNA sequencing of spleen monocytes flow-sorted during the acute phase of first or second infection (7 days p.i.). In each case, the number of differentially expressed genes (DEG, $p_{adj}$ <0.01, >1.5-fold change) was assessed relative to uninfected controls. Pie shows the proportion of DEG unique to second infection (i.e. not shared with first infection). (**B–C**) Log2 fold change (FC) of (**B**) signature genes used to predict monocyte differentiation into inflammatory macrophages (MΦ) or monocyte-derived dendritic cells (DC); and (**C**) chemokines, cytokines, and their receptors. Log2FC is shown relative to uninfected controls. (**D**) Plasma concentration of CXCL10 and IFNɣ during the acute phase of first (n = 4) or second (n = 5) infection (box-plots show median and IQR). Note that both plasma proteins were below the limit of detection in uninfected controls. (**E**) Log2FC of signature genes used to predict monocyte differentiation towards alternative fates (log2FC is shown relative to uninfected controls). In (**A–C** and **E**), n = 5 for infected mice and n = 6–7 for uninfected controls. Square brackets indicate that common gene names were used.

The online version of this article includes the following figure supplement(s) for figure 4:

**Figure supplement 1.** Monocytes minimise inflammation in tolerised hosts.

We therefore explored alternative monocyte fates, such as those associated with immune regulation, wound healing, and tissue repair. However, spleen monocytes were not polarised towards alternatively activated macrophages and nor did they induce signature genes associated with myeloid-derived suppressor cells (*Gabrilovich, 2017*) or reparative Ly6C[lo] monocytes (*Jung et al., 2017*; *Figure 4E*). Furthermore, they did not upregulate anti-inflammatory cytokines or checkpoint inhibitors (*Figure 4C* and *Figure 4—figure supplement 1C*). To make sense of their complex transcriptional profile we therefore turned once more to clueGO. In contrast to first infection, we found minimal enrichment of GO terms linked to host defence; instead, all major supergroups in second infection related to regulation of cell cycle and nuclear division (*Figure 5A–C*). This localised

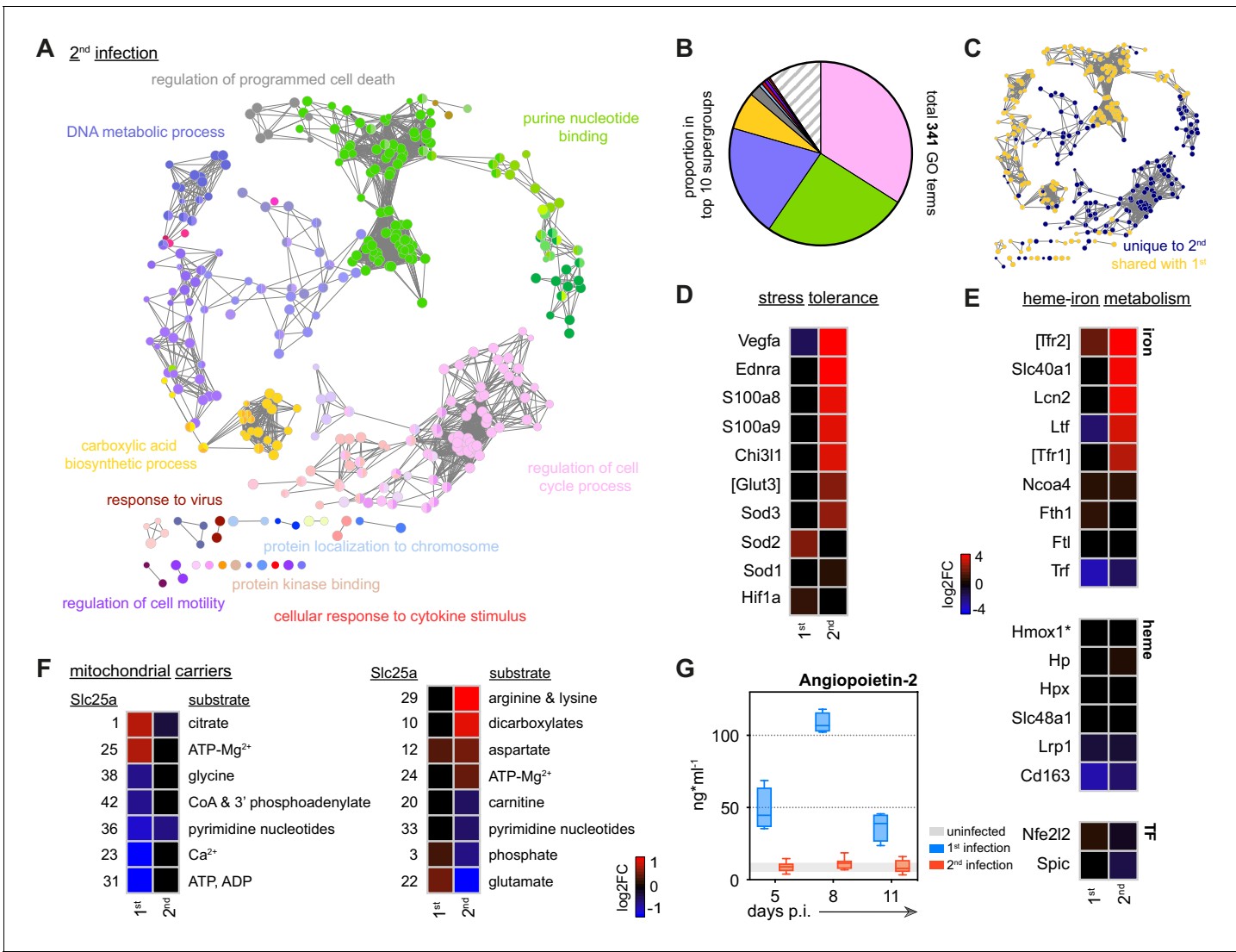

**Figure 5.** Disease tolerance is established after one malaria episode. (A and B) ClueGO network of DEG in spleen monocytes during the acute phase of second infection. Each node represents a significantly enriched gene ontology (GO) term and node size is determined by $p_{adj}$. Related GO terms that share >40% of genes are connected by a line and organised into functional groups (each given a unique colour). Supergroups are formed when GO terms are shared between more than one group. The names of the top 10 (super)groups (lowest $p_{adj}$) are displayed and (B) shows their proportion of total GO terms. (C) Recoloured clueGO network indicating the GO terms that are unique to second infection (or shared with first infection). (D–F) Log2 fold change (FC) of genes that (D) promote stress tolerance; (E) regulate heme-iron metabolism; and (F) encode mitochondrial carrier proteins. Log2FC is shown relative to uninfected controls. (G) Plasma concentration of Angiopoietin-2 during the acute phase of first (n = 4) or second (n = 8) infection (box-plots show median and IQR). The grey shaded area represents min. to max. measurements in uninfected controls (n = 3). In (A–F), n = 5 for infected mice and n = 6–7 for uninfected controls. Square brackets indicate that common gene names were used.

proliferation of monocytes may support a critical spleen response without having to engage increased production and recruitment from the bone marrow.

So what is the function of spleen monocytes in tolerised hosts? First off, they upregulate the expression of two alarmins (*S100a8* and *S100a9*) (*Figure 5D*) that form the heterodimer calprotectin; this acts as an endogenous TLR4 ligand to silence the inflammatory response to pathogen-derived or damage-associated signals. Notably, increased expression of these alarmins in early life prevents hyperinflammation to protect neonates from sepsis (*Ulas et al., 2017*). Similarly, *Chi3l1* (encoding the prototypical chitinase-like protein) is also upregulated in second infection and can attenuate inflammasome activation to minimise collateral tissue damage (*Dela Cruz et al., 2012*). In both cases, these mechanisms promote host control of inflammation without impairing pathogen clearance.

Monocytes also appear to minimise the effects of hypoxia and oxidative stress on their environment. For example, in first infection monocytes upregulate *Sod2*, which encodes a mitochondrial protein required to safeguard inflammatory macrophages from the reactive oxygen species that they produce. But in second infection they instead upregulate *Sod3*, which can be secreted to inactivate extracellular reactive oxygen species in the surrounding tissue (*Yao et al., 2010*; *Figure 5D*). In much the same way, monocytes minimise host stress by upregulating *Ednra*, which can scavenge endothelin-1 to prevent vasoconstriction and hypertension (*Czopek et al., 2019*), and they increase expression of *Vegfa* to promote angiogenesis. Evidently, the transcriptional profile of spleen monocytes suggests that they take on a tissue protective role in second infection.

In support of this argument, monocytes differentially regulate heme-iron metabolism to deal with the release of toxic free heme and reactive iron ($Fe^{2+}$) from ruptured red cells (*Figure 5E*). In brief, they increase their transcription of a core set of genes required to sequester extracellular iron (*Ltf* encoding lactotransferrin), import sequestered iron for detoxification (*Tfr1/Tfr2* encoding the transferrin receptors) and then export detoxified iron for use in the production of new red cells (*Slc40a1* encoding ferroportin). Notably, monocytes do not appear to increase their iron storage capacity, which has been associated with tissue damage in malaria (*Gozzelino et al., 2012*). And nor do they upregulate expression of heme oxygenase-1 (*Hmox1*), which is required to detoxify free heme. These data therefore indicate that monocytes specifically enhance iron recycling to promote tolerance to hemolysis – a major source of stress in malaria. Crucially, this only occurs during reinfection, despite an identical red cell loss in naive and tolerised hosts (*Figure 3C*).

Taken together, these data clearly show that spleen monocytes initiate a transcriptional programme designed to promote tolerance to malaria parasites upon reinfection. This is achieved in two ways – first by minimising inflammation to reduce collateral tissue damage and second by engaging pathways that can impart stress tolerance on their environment. Significantly, mice were first infected with the avirulent parasite genotype AS, suggesting that parasite-derived signals may be sufficient to redirect monocyte fate.

## Metabolic rewiring underpins monocyte fate

We moved on to ask whether the fate and function of monocytes could be underpinned by metabolic reprogramming. Cellular metabolism has emerged as a key determinant of monocyte-to-macrophage differentiation (*O'Neill et al., 2016*) and so we looked again at transcriptional control of the key enzymes involved in glycolysis and oxidative phosphorylation. We found that both pathways were comparably induced during first and second infection with one notable exception: monocytes switched from upregulating the glucose transporter *Slc2a1* in first infection to *Slc2a3* in second infection (*Figure 2—figure supplement 1C–D* and *Figure 4—figure supplement 1D–E*). *Slc2a3* encodes the facilitative GLUT3 transporter, which unlike most other transporters can continue to import glucose under hypoglycaemic conditions (*Simpson et al., 2008*). Switching to GLUT3 may therefore constitute a cell-intrinsic adaptation that allows monocytes to tolerate infection-induced stress. When we looked deeper into the transcriptional control of cell metabolism we found that monocytes also differentially expressed their mitochondrial carrier proteins (*Figure 5F*). These membrane-embedded proteins provide the cellular wiring that connects metabolic reactions in the cytosol with the mitochondrial matrix by transporting metabolites, nucleotides, and co-enzymes across the inner mitochondrial membrane (*Palmieri, 2013*); in this way, mitochondrial carrier proteins facilitate the complex crosstalk between all major metabolic pathways. In first infection, spleen monocytes primarily upregulate a carrier protein (encoded by *Slc25a1*) whose major substrate is citrate, a metabolite

known to accumulate in inflammatory macrophages. In contrast, monocytes upregulate *Slc25a29* during reinfection and this carrier protein shuttles arginine, which can be fluxed through the arginase pathway to promote tolerance and wound healing (*O'Neill et al., 2016*). By re-wiring their mitochondria, monocytes may thus be enabling their specialised tissue protective functions in tolerised hosts.

## Disease tolerance is established after one malaria episode

Host control of inflammation may provide a quick and effective way to establish disease tolerance (*Medzhitov et al., 2012*); so far, we have shown that inflammation is minimised in the bone marrow (preventing emergency myelopoiesis), blood (decreasing plasma interferon), and spleen (diverting monocyte fate). To directly show that this coincides with a reduction in pathology we measured circulating levels of Angiopoietin-2. This vascular growth factor is the best biomarker of endothelium activation and dysfunction in human malaria and the most accurate prognostic marker of mortality in children (*Yeo et al., 2008*). We found that in first infection Angiopoietin-2 levels increased by more than an order of magnitude but in second infection – with an identical parasite burden – levels did not deviate from a healthy uninfected baseline (*Figure 5G*). A single malaria episode can therefore induce host adaptations that promote disease tolerance and provide long-lived clinical immunity.

## Malaria does not induce epigenetic reprograming of bone marrow monocytes

So how do tolerised hosts control inflammation independently of pathogen load? To begin to answer this question, we looked once again at the functional specialisation of monocytes in second infection. Our data are consistent with a model of innate memory, whereby myeloid progenitors in the bone marrow are epigenetically reprogrammed during first infection to intrinsically modify the response of monocytes to reinfection. To test this hypothesis, we asked if malaria induces heritable histone modifications that alter the epigenetic landscape of inflammatory monocytes before their release from the bone marrow. And since tolerance can persist in the absence of parasitaemia, we isolated monocytes from once-infected mice one month after drug cure (day 70, see *Figure 1A*). Crucially, this was exactly the same time-point at which we had performed all of our reinfection studies (*Figure 3A*). In this experiment, we interrogated the distribution of histone modifications genome-wide using ChIPseq and asked whether (i) transcription start sites were marked with H3K27ac to activate transcription (ii) enhancers or superenhancers were marked with H3K4me1 to promote gene expression or (iii) DNA was condensed into heterochromatin by H3K9me3 to silence gene expression.

We used the motif discovery software HOMER (*Heinz et al., 2010*) to identify peaks and visualised peaks with the genomics exploration tool Integrative Genomics Viewer (*Thorvaldsdóttir et al., 2013*). In the first instance, we looked at the histone modification profiles of genes that define monocyte function in first and second infection; for example, genes associated with inflammation versus proliferation. Subscribing to the notion that ChIPseq reveals qualitative (not quantitative) differences (*Ma et al., 2018*; *Orlando et al., 2014*), we simply asked whether these genes were marked or not marked. Our prediction was that genes that were upregulated during first infection but silenced during reinfection (tolerised genes) would lose marks associated with active transcription or be condensed into inactive heterochromatin. Conversely, specialised genes (those upregulated exclusively during second infection) would gain marks to promote transcription. Remarkably however, we found that in almost every case the histone modification profiles of tolerised and specialised genes were identical between monocytes isolated from once-infected mice and uninfected controls (*Figure 6A–B* and *Figure 6—figure supplement 1*).

Even when we used HOMER to call differentially modified regions (DMR) and quantify differences between once-infected and control mice, we found little evidence to support epigenetic reprogramming of monocytes – of the 2848 tolerised/specialised genes identified by RNAseq 95% had no detectable histone modifications (*Figure 6C*). And in those rare cases where a DMR was called, HOMER assigned a low confidence peak score (*Supplementary file 2*). Innate memory can not therefore easily explain the widespread transcriptional changes that lead to the functional specialisation of monocytes in tolerised hosts.

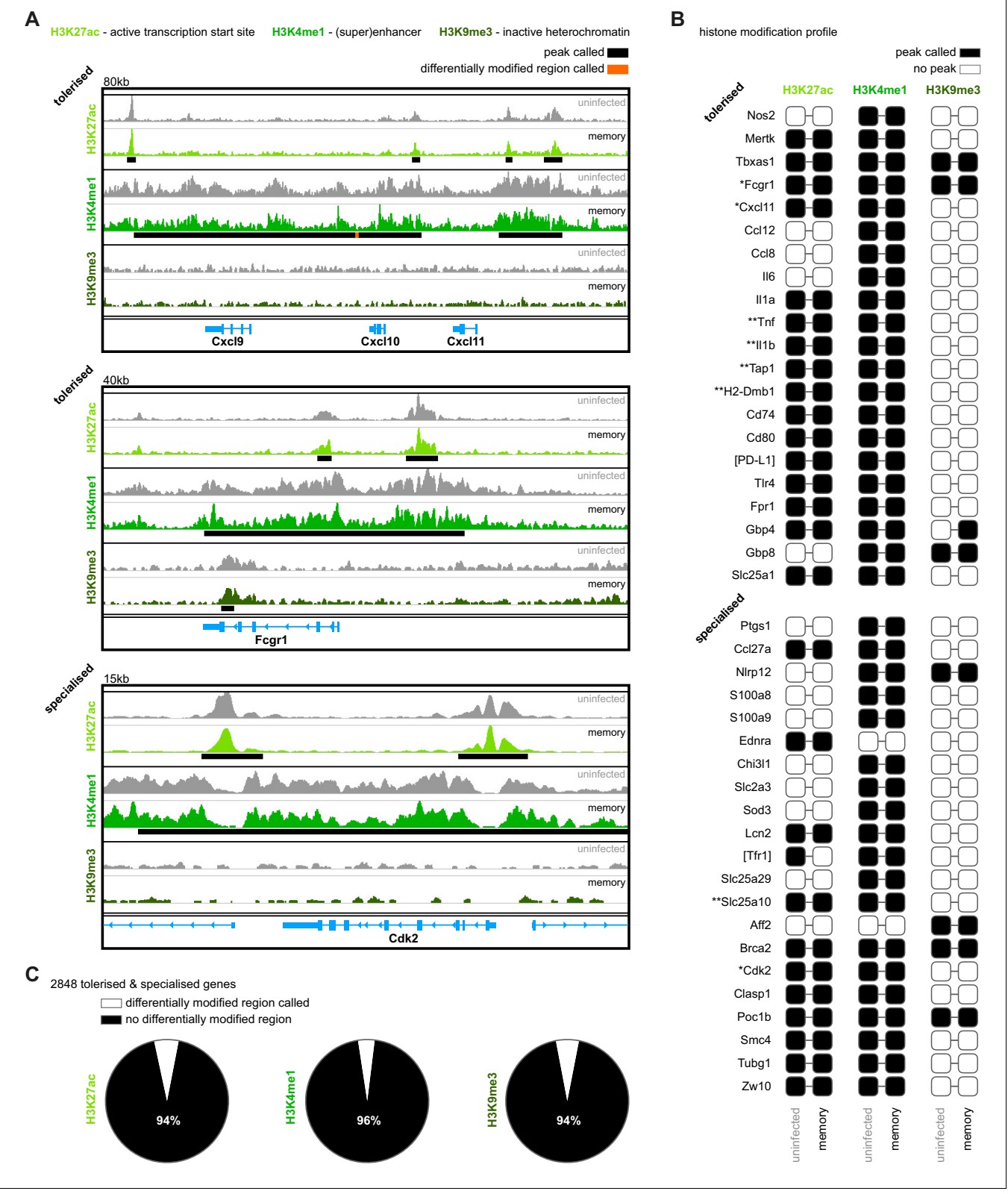

**Figure 6.** Malaria does not induce epigenetic reprogramming of bone marrow monocytes. (**A**) Chromatin immunoprecipitation (ChIP)seq of bone marrow monocytes flow-sorted from once-infected mice (AJ, memory, 70 days p.i.) and uninfected controls. Shown are the Integrative Genomics Viewer (IGV) traces (autoscaled) of three loci encoding genes that are transcriptionally tolerised (upregulated in first but not second infection) or specialised (upregulated only in second infection). Peaks were called relative to non-immunoprecipitated input DNA and are shown for uninfected controls (black

*Figure 6 continued on next page*

*Figure 6 continued*

line). Regions of the genome that were differentially marked between once-infected mice and uninfected controls are underlined in orange (differentially modified regions). (B) Histone modification profiles of bone marrow monocytes from once-infected mice and uninfected controls. Black squares indicate that a peak was called within 10 kb (H3K27ac and H3K9me3) or 100 kb (H3K4me1) of the transcription start site. Square brackets indicate that common gene names were used. IGV traces are shown in (A) and *Figure 6—figure supplement 1* for genes marked with one or two asterisks, respectively. (C) Pies show the proportion of tolerised/specialised genes (n = 2848) annotated with or without a differentially modified region (annotated to the nearest gene). In (A–C), the data shown are pooled from independent biological replicates (see Materials and methods).

The online version of this article includes the following figure supplement(s) for figure 6:

**Figure supplement 1.** Malaria does not induce epigenetic reprograming of bone marrow monocytes.

## Monocytes are transcriptionally reprogrammed in the remodelled spleen

An alternative explanation is that monocyte fate is imprinted within the spleen; after all, tissue printing is a key route to organ-specific identity during monocyte to macrophage differentiation (*Scott et al., 2016*; *van de Laar et al., 2016*). We therefore looked for transcriptional evidence of long-lived changes in spleen monocytes that can persist after parasite clearance. We found 111 differentially expressed genes in monocytes isolated from once-infected mice compared to uninfected controls; remarkably, most of these genes were not differentially expressed during acute or chronic infection. Instead, this transcriptional signature was unique to the memory phase and was further enhanced upon reinfection (*Figure 7A*). This included the transcription factor *Maf*, which regulates macrophage programming in vivo (*Kang et al., 2017*; *Liu et al., 2020*), and *Sirpa*, which regulates recognition of self (*Bian et al., 2016*). The majority of genes, however, related to cell cycle and nuclear division – specialised functions of spleen monocytes in tolerised hosts. Indeed, we found remarkable overlap in the top GO terms identified in memory and second infection (*Figure 7B*). A critical part of the transcriptional programme designed to promote tolerance to malaria parasites is therefore already engaged in monocytes prior to reinfection. And this transcriptional signature does not appear to require epigenetic reprogramming in the bone marrow. These data thus provide compelling evidence that malaria may remodel the spleen to imprint tolerance.

## Discussion

In this study, we show that mechanisms of disease tolerance can persist after pathogen clearance to minimise tissue damage, stress and pathology during subsequent infections. These inducible mechanisms of tolerance therefore provide memory and constitute an alternative strategy of acquired immunity. Crucially, these adaptations function to preserve key homeostatic processes and protect life, and are therefore likely to be the primary form of host defense when sterile immunity can not be generated. This is particularly relevant in malaria, where even partial control of parasite densities is not usually demonstrable until adolescence (*Marsh and Kinyanjui, 2006*). It seems likely that many complementary strategies of tolerance will need to cooperate to provide protection and we identify three mechanisms in the myeloid compartment alone – emergency myelopoiesis is disarmed; tissue-resident macrophages become resistant to malaria-induced ablation; and spleen monocytes eschew an inflammatory macrophage fate to take on a tissue protective role. These adaptations operate independently of pathogen load and coincide with a reduction in systemic inflammation; what's more, tissue integrity is preserved, ferric iron stores are maintained and endothelium activation/dysfunction is avoided during reinfection. Acquired mechanisms of disease tolerance can therefore avert hallmark features of severe malaria.

Host control of inflammation thus appears to provide an effective route to disease tolerance and monocyte activation is clearly not hardwired. However, their functional specialisation in tolerised hosts is not underpinned by epigenetic reprogramming of progenitor cells in the bone marrow but instead seems to be imprinted within the remodelled spleen. Tissue printing of immune cell function is well recognised, and allows monocytes to take on a remarkably diverse range of organ-specific roles. This has been best characterised during the differentiation of monocytes into long-lived tissue resident macrophages, in which local signals imprint specialised functions that are unique to every tissue (*Scott et al., 2016*; *van de Laar et al., 2016*). In this way, tissue printing maximises the

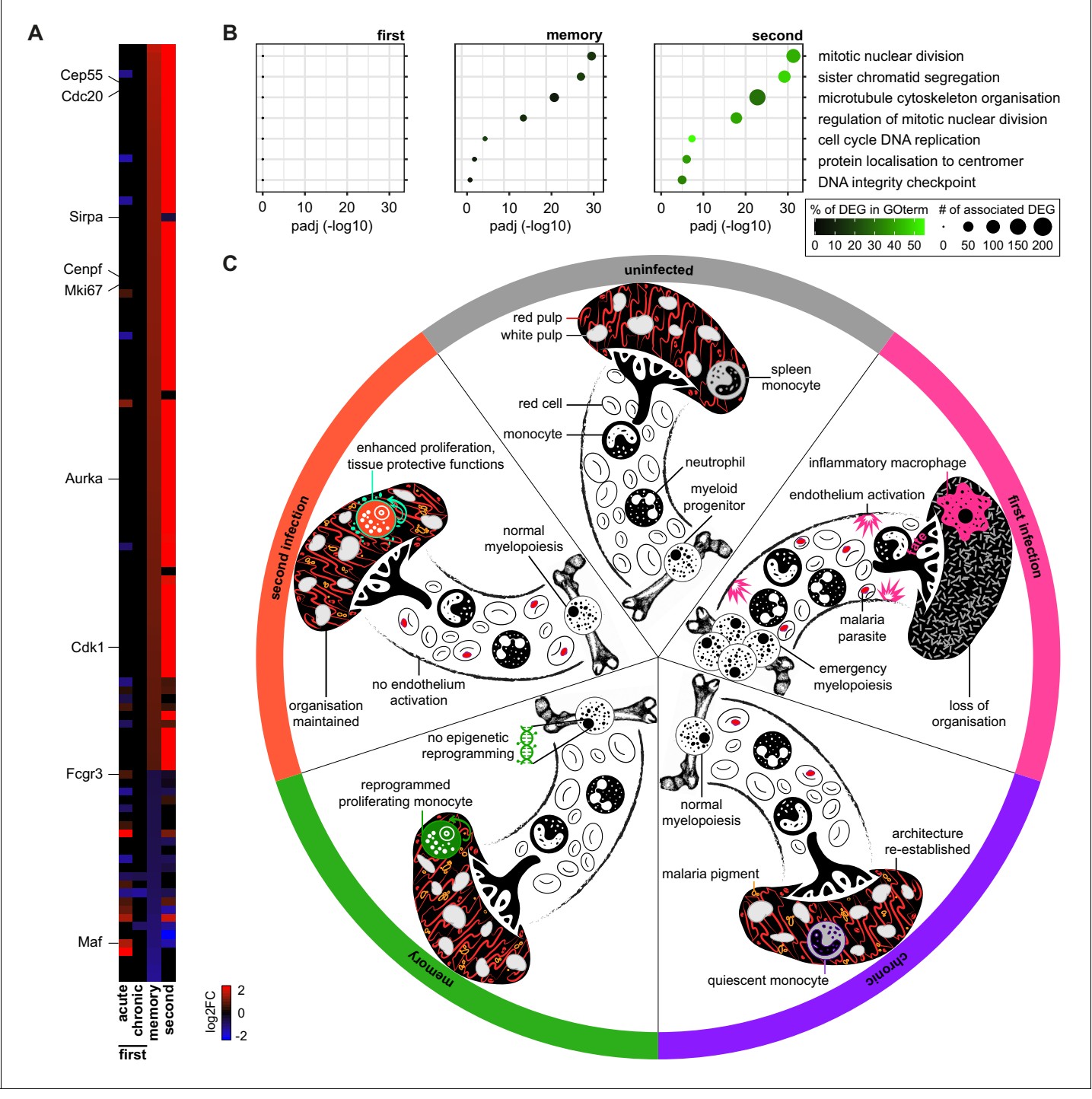

**Figure 7.** Monocytes are transcriptionally reprogrammed in the remodelled spleen. (**A**) RNA sequencing of spleen monocytes flow-sorted from AJ-infected mice (7 and 40 days p.i. for acute and chronic, respectively), once-infected mice (memory, 70 days p.i.), and reinfected mice (acute, 7 days p.i.). The heatmap shows all 111 differentially expressed genes in once-infected mice (DEG, relative to uninfected controls, $p_{adj}$ <0.01, >1.5-fold change). (**B**) GO analysis of DEG in spleen monocytes during first infection (7 days p.i.), memory phase (70 days p.i.) and second infection (7 days p.i.). Mice were infected with *P. chabaudi* AJ and the top GO terms in once-infected mice are shown. (**C**) Working model of disease tolerance in malaria, showing the major changes in the myeloid compartment throughout chronic infection, convalescence and reinfection. Note that in the memory phase (one month after drug cure) there is no evidence that monocytes are epigenetically reprogrammed in the bone marrow but they are transcriptionally reprogrammed in the spleen. We therefore propose that the remodelled spleen imprints monocytes with tissue protective functions. In (**A** and **B**) n = 5–6 for infected mice and n = 6–7 for uninfected controls. (**C**) Icons credit: https://thenounproject.com/.

The online version of this article includes the following figure supplement(s) for figure 7:

*Figure 7 continued on next page*

*Figure 7 continued*

**Figure supplement 1.** Tissue printing shapes the transcriptional programme of recruited monocytes to meet the needs of the niche.

plasticity of myeloid cells. One of many challenges when resolving the acute phase response to malaria is to refill the red pulp macrophage niche, which was obliterated by acute infection; this is achieved through the recruitment and differentiation of bone marrow-derived monocytes (*Lai et al., 2018*). We find that red pulp macrophages isolated from once-infected mice are transcriptionally identical to those from uninfected controls (*Figure 7—figure supplement 1*), which shows that tissue printing can precisely match the transcriptional programme of recruited monocytes with the functional requirements of the tissue – in this case, to re-establish ferric iron stores. It is clear then that the tissue niche imprints organ-specific identity onto monocytes that take residence to maintain homeostatic processes. But similar mechanisms must also operate to control the fate of monocytes that do not take residence and instead are recruited to deal with injury or infection (*Guilliams et al., 2018*). In this scenario, how might the tissue niche be remodelled by malaria to promote tolerance?

A niche is often considered to be a small self-contained tissue scaffold that provides the physical structure required to target specialised signals to resident cells or cells transiting the niche. Nevertheless, a niche can also be a dynamic and expansive space, such as the open circulatory system created by the pulp cords and sinuses of the spleen (*Guilliams et al., 2020*). Malaria causes pronounced splenomegaly and disruption of architecture, including a complete loss of boundaries between red and white pulp. During chronic infection, splenomegaly begins to resolve and these boundaries are re-established but the spleen may not be put back exactly as it was before – it may acquire new and altered features. For example, malaria pigment accumulates throughout the red pulp, and these persistent insoluble deposits may regulate the activation and differentiation of monocytes arriving in the remodelled spleen. In support of this idea, malaria pigment can suppress the oxidative burst (*Schwarzer et al., 1992*) and interferon-induced upregulation of class II MHC (*Schwarzer et al., 1998*) in human monocytes in vitro. Alternatively, re-assembly of the stromal cell networks that compartmentalise the spleen may alter cell patterning and thereby change the distribution and density of key signals that control monocyte fate (*Bonnardel et al., 2019*; *Koliaraki et al., 2020*). Or perhaps the tissue niche required to promote tolerance is already present in the spleen (even in a naive host) but is lost when the spleen becomes enlarged and disorganised in first infection. In this example, it is the preservation of spleen architecture that facilitates the functional specialisation of monocytes in tolerised hosts. In every scenario, monocytes arriving in the remodelled spleen receive a new combination of signals that can minimise inflammation and induce their tissue protective functions (see *Figure 7C* for a working model). Importantly, tissue remodelling and the accumulation of malaria pigment is observed in the human spleen (*Buffet et al., 2011*), and we therefore suggest that this is a conserved strategy through which malaria imprints tolerance. This will almost certainly benefit the parasite as well as the host; after all, if you can minimise host sickness behaviour you will likely maximise the probability of onward transmission.

Other changes to the niche include the appearance of malaria-specific memory B and T cells, which provide effective adaptive immunity in our reinfection model (peak AJ parasitaemia is reduced from 25% in naive hosts to 0.5% in tolerised hosts). Nonetheless, this relatively low pathogen load is more than sufficient to promote the differentiation of spleen monocytes into inflammatory macrophages, as demonstrated in naive hosts infected with *P. chabaudi* AS. So how might adaptive mechanisms of immunity influence the myeloid response independently of pathogen load? One route through which B cells may regulate monocyte fate is through the production of opsonising antibodies, which recognise merozoites and/or infected red cells. Indeed, it has been shown that phagocytosis of immune complexes in the context of Toll-like receptor signaling can induce M2 (or regulatory) macrophages (*Guilliams et al., 2014*). However, in these experiments M2 macrophages increased production of the pyrogenic cytokines IL-1β, IL-6 and TNF, and malaria-specific opsonising antibodies have been directly shown to promote inflammation in human macrophages (*Osier et al., 2014*; *Zhou et al., 2012*). Instead, the generation of memory T cells that reduce the production of IFNɣ may be a more simple explanation; in agreement, we find that circulating levels of IFNɣ are substantially reduced in second infection. Nevertheless, signaling through the IFNɣ receptor is not a prerequisite for the differentiation of inflammatory macrophages (*Xue et al., 2014*) and the early source of

IFNγ in naive hosts is not T cell-derived (*Artavanis-Tsakonas and Riley, 2002*). Changes in cytokine production may therefore need to be much more extensive if T cells are to push monocytes away from an inflammatory fate. In this context, it is notable that malaria-specific memory T cells have been shown to adopt new effector functions upon recall (*Zander et al., 2017*); as such, they may be well placed to modify the availability of diverse signals within the tissue niche.

Future studies should therefore aim to uncouple the relative contribution of T cell signals versus structural changes (inc the appearance of malaria pigment) in redirecting monocyte fate. This may be achieved by transplanting spleens from tolerised to naive hosts and then infecting with the avirulent parasite genotype AS. In this setting, it will be possible to a) provide definitive evidence that monocyte fate is imprinted within the spleen and b) assess the role of memory T cells by injecting donor mice with a depleting anti-CD3 antibody prior to transplantation. Interpretation of these experiments may be complicated however by the fact that recipient mice would still be expected to trigger emergency myelopoiesis and lose their tissue resident macrophages. Systemic inflammation (and the disruption of homeostatic processes) may override any local signals in the spleen that favour tolerance – reinfection is characterised by diverse adaptations in distinct tissues after all, and presumably these will need to work in unison to provide clinical immunity.

It is therefore important that we consider how other host adaptations might be established. Emergency myelopoiesis can be disarmed in the bone marrow; how is this avoided second time around despite an identical pathogen load? It could be that changes to the tissue niche promote tolerance, in much the same way as described in the spleen, as first infection leads to a loss of cellularity and tissue integrity in the bone marrow, which inevitably requires repair. But it is also possible that cell-intrinsic changes establish tolerance in this case. Current evidence indicates that systemic inflammation, which can push haematopoietic stem cells (HSC) down the myeloid path (*Mitroulis et al., 2018*), is triggered in the bone marrow by plasmacytoid dendritic cells (pDC), which produce type I interferon (*Spaulding et al., 2016*). Can you intrinsically modify pDC to attenuate their function (e.g. by increasing their activation threshold)? Or limit their access to parasites and their pyrogenic products? Or perhaps tolerance operates further upstream – the expression of pattern recognition receptors on HSC could be restricted to prevent their direct activation by malaria parasites. Whatever the mechanism, we believe that disarming emergency myelopoiesis in the bone marrow is just as important as imprinting monocytes with tissue protective functions in the spleen.

We also show that long-lived prenatally seeded tissue-resident macrophages become resistant to malaria-induced ablation. Tissue-resident macrophages were often considered to be immune sentinels that provide a first line of defence against invading pathogens but actually they have been shown time and again to be sensitive to stress, and their disappearance in response to injury or infection is well recognised (*Aegerter et al., 2020*; *Blériot et al., 2015*; *Machiels et al., 2017*; *Scott et al., 2016*). Their functions instead mainly relate to development, tissue repair, and homeostasis. So how does malaria cause the death of tissue-resident macrophages in naive hosts and why are they protected in tolerised hosts? We suggest that the release and catabolism of free heme from ruptured red cells can explain this phenomenon. Free heme is toxic to tissue-resident macrophages (*Cambos and Scorza, 2011*; *Ferreira et al., 2008*; *Haldar et al., 2014*; *Theurl et al., 2016*) and the detoxification of free heme is known to confer disease tolerance (*Seixas et al., 2009*). In our model, heme accumulation is likely to be similar between first and second infection as parasite burden and red cell loss are comparable. The preservation of tissue-resident macrophages could therefore be a consequence of more efficiently detoxifying free heme to reduce its bioavailability and pathogenicity. Since we find no evidence that monocytes transcriptionally regulate their capacity to metabolise heme during reinfection (they exclusively upregulate iron recycling) other cell types would need to carry out this role, including hepatocytes and renal proximal tubule epithelial cells (*Ramos et al., 2019*; *Theurl et al., 2016*). In this scenario, metabolic adaptations that operate in non-lymphoid tissues must also be inducible, and would persist after pathogen clearance to provide memory and clinical immunity.

Acquired immunity to malaria is often considered to be slow and ineffective – but this is only true if we consider resistance (the ability to eliminate parasites) in isolation. Despite being an equally important part of host defense, the central role of disease tolerance in acquired immunity to malaria has not been fully appreciated. We demonstrate that disease tolerance can be attained after a single malaria episode and can persist in the absence of parasitaemia to allow the host to tolerate subsequent infections. We therefore argue that mechanisms of disease tolerance provide an alternative

strategy of acquired immunity that functions not to kill parasites but to limit the damage that they can cause. Adaptations in the myeloid compartment minimise inflammation and promote stress tolerance, and it is likely that these will need to be complemented by adaptations in other immune compartments such as memory T cell reservoirs, which orchestrate innate and adaptive immunity. Furthermore, changes to the immune response will need to be complemented by long-lived metabolic adaptations that maintain homeostasis and preserve organ function. Together, all of these adaptations can work in concert to protect host tissues and minimise pathology. The mechanisms of disease tolerance uncovered in this study are therefore likely just the tip of the iceberg but they can begin to explain how children in endemic areas acquire immunity to severe malaria so quickly and without the need for improved pathogen control (*Gonçalves et al., 2014*; *Gupta et al., 1999*; *Marsh and Snow, 1999*).

# Materials and methods

## Key resources table

| Reagent type (species) or resource | Designation | Source or reference | Identifiers | Additional information |
|---|---|---|---|---|
| Strain, strain background (*Mus musculus* C57Bl/6J female) | C57Bl/6 | *The Jackson Laboratory* | RRID:IMSR_JAX:000664 | Bred and housed in individually ventilated cages (SPF conditions) at the University of Edinburgh |
| Strain, strain background (*Plasmodium chabaudi chabaudi* AS) | *P. chabaudi* AS | *The European malaria reagent repository* http://www.malariaresearch.eu | Clone 28AS11 | |
| Strain, strain background (*Plasmodium chabaudi chabaudi* AJ) | *P. chabaudi* AJ | | Clone 96AJ15 | |
| Strain, strain background (*Anopheles stephensi* SD500) | Mosquitoes | Reared in-house at the University of Edinburgh | | |
| Antibody | Anti-mouse B220 (rat monoclonal) | Clone RA3-6B2 *eBioscience* - sold by *ThermoFisher* | RRID:AB_10717389 | (0.2 µl) per test = 2 million cells in 100 µl volume |
| Antibody | Anti-mouse CD3ε (Armenian hamster monoclonal) | Clone 145–2 C11 *BioLegend* | RRID:AB_312676 | (0.3 µl) per test |
| Antibody | Anti-mouse CD4 (rat monoclonal) | Clone RM4-5 *BioLegend* | RRID:AB_312718 | (0.3 µl) per test |
| Antibody | Anti-mouse CD8a (rat monoclonal) | Clone 53–6.7 *BioLegend* | RRID:AB_312750 | (0.3 µl) per test |
| Antibody | Anti-mouse CD11b (rat monoclonal) | Clone M1/70 *BioLegend* | RRID:AB_312798 | (0.1 µl) per test |
| Antibody | Anti-mouse CD11c (Armenian hamster monoclonal) | Clone N418 *BioLegend* | RRID:AB_313776 | (0.15 µl) per test |
| Antibody | Anti-mouse CD16/32 (rat monoclonal) | Clone 93 *eBioscience* - sold by *ThermoFisher* | RRID:AB_469598 | (0.5 µl) per test |
| Antibody | TruStain FcX anti-mouse CD16/32 (rat monoclonal) | Clone 93 *BioLegend* | RRID:AB_1574973 | (2 µl) per test blocks FcγR II/III prior to antibody staining |

*Continued on next page*

Continued

| Reagent type (species) or resource | Designation | Source or reference | Identifiers | Additional information |
|---|---|---|---|---|
| Antibody | Anti-mouse CD19 (rat monoclonal) | Clone 6D5 *BioLegend* | RRID:AB_313646 | (0.1 µl) per test |
| Antibody | Anti-mouse CD27 (Armenian hamster monoclonal) | Clone LG.7F9 *eBioscience - sold by ThermoFisher* | RRID:AB_465614 | (0.3 µl) per test |
| Antibody | Anti-mouse CD34 (rat monoclonal) | Clone RAM34 *eBioscience - sold by ThermoFisher* | RRID:AB_465021 | (0.4 µl) per test |
| Antibody | Anti-mouse CD71 (rat monoclonal) | Clone RI7217 *BioLegend* | RRID:AB_10899739 | (0.3 µl) per test |
| Antibody | Anti-mouse CD115/Csf1r (rat monoclonal) | Clone AFS98 *BioLegend* | RRID:AB_2562760 | (0.3 µl) per test |
| Antibody | Anti-mouse CD135/Flt3 (rat monoclonal) | Clone A2F10 *eBioscience - sold by ThermoFisher* | RRID:AB_465859 | (2.5 µl) per test |
| Antibody | Anti-mouse CD169 (rat monoclonal) | Clone 3D6.112 *BioLegend* | RRID:AB_2563910 | (1 µl) per test |
| Antibody | Anti-mouse cKit/CD117 (rat monoclonal) | Clone 2B8 *eBioscience - sold by ThermoFisher* | RRID:AB_1834421 | (0.3 µl) per test |
| Antibody | Anti-mouse CX3CR1 (mouse monoclonal) | Clone SA011F11 *BioLegend* | RRID:AB_2564493 | (0.3 µl) per test |
| Antibody | Anti-mouse F4/80 (rat monoclonal) | Clone BM8 *BioLegend* | RRID:AB_10901171 | (0.8 µl) per test |
| Antibody | Anti-mouse IAb (mouse monoclonal) | Clone AF6-120.1 *BioLegend* | RRID:AB_313724 | (0.5 µl) per test |
| Antibody | Anti-mouse Ly6C (rat monoclonal) | Clone HK1.4 *BioLegend* | RRID:AB_2562177 | (0.1 µl) per test |
| Antibody | Anti-mouse Ly6G (rat monoclonal) | Clone 1A8-Ly6g *eBioscience - sold by ThermoFisher* | RRID:AB_2573893 | (0.2 µl) per test |
| Antibody | Anti-mouse NK1.1 (mouse monoclonal) | Clone PK136 *BioLegend* | RRID:AB_313396 | (0.3 µl) per test |
| Antibody | Anti-mouse Nr4a1/Nur77 (mouse monoclonal) | Clone 12.14 *eBioscience - sold by ThermoFisher* | RRID:AB_1257209 | (0.3 µl) per test intracellular stain |
| Antibody | Anti-mouse Sca1/Ly6a (rat monoclonal) | Clone D7 *BioLegend* | RRID:AB_2562275 | (2 µl) per test |
| Antibody | Anti-mouse Ter119 (rat monoclonal) | Clone Ter119 *BioLegend* | RRID:AB_313712 | (0.3 µl) per test |
| Antibody | Anti-mouse VCAM-1 (rat monoclonal) | Clone 429 *BioLegend* | RRID:AB_1595594 | (0.5 µl) per test |
| Antibody | Anti-H3K27ac ChIPseq grade (rabbit polyclonal) | *Diagenode* #C15410196 see our optimised ChIPseq protocol dx.doi.org/10.17504/ protocols.io.bja3kign | RRID:AB_2637079 | (2 µg) per ChIP |
| Antibody | Anti-H3K4me1 ChIPseq grade (rabbit polyclonal) | *Diagenode* #C15410037 see our optimised ChIPseq protocol dx.doi.org/10.17504/ protocols.io.bja3kign | RRID:AB_2561054 | (5 µg) per ChIP |

*Continued on next page*

*Continued*

| Reagent type (species) or resource | Designation | Source or reference | Identifiers | Additional information |
|---|---|---|---|---|
| Antibody | Anti-H3K9me3 ChIPseq grade (rabbit polyclonal) | *Diagenode* #C15410193 see our optimised ChIPseq protocol dx.doi.org/10.17504/ protocols.io.bja3kign | RRID:AB_2616044 | (1 µg) per ChIP |
| Sequence-based reagent | Forward primer | 5-GCGAGAAAGT TAAAAGAATTGA-3 | | For measuring *P. chabaudi* blood-stage parasitaemia by quantitative PCR |
| Sequence-based reagent | Reverse primer | 5-CTAGTGAGT TTCCCCGTGTT-3 | | |
| Sequence-based reagent | Probe | [6FAM] - AAATTAAGCC GCAAGCTCCACG - [TAM] | | |
| Commercial assay or kit | Quick DNA Universal Microprep Kit | *Zymo Research* | D4074 | |
| Commercial assay or kit | IFNγ mouse ELISA kit, extra sensitive | *Invitrogen* - sold by *ThermoFisher* | BMS609 | |
| Commercial assay or kit | IP-10 (CXCL10) mouse ELISA kit | *Invitrogen* - sold by *ThermoFisher* | BMS6018 | |
| Commercial assay or kit | Mouse/rat Angiopoietin-2 quantine ELISA kit | *R&D Systems* | MANG20 | |
| Commercial assay or kit | Foxp3 / Transcription Factor Staining Buffer Set | *eBioscience* - sold by *ThermoFisher* | 00-5523-00 | |
| Commercial assay or kit | SMART-Seq v4 Ultra Low Input RNA Kit | *Takara Bio* | 634891 | |
| Commercial assay or kit | Nextera XT DNA Library Preparation Kit | *Illumina* | FC-131-1024 | |
| Commercial assay or kit | True MicroChIP kit | *Diagenode* | C01010130 | |
| Commercial assay or kit | MicroPlex Library Preparation Kit v2 | *Diagenode* | C05010012 | |
| Commercial assay or kit | RNA Clean and Concentrator-5 Kit | *Zymo Research* | R1013 | |
| Commercial assay or kit | GeneChip WT Pico Kit | *Affymetrix* - sold by *ThermoFisher* | 902622 | |
| Commercial assay or kit | GeneChip Mouse Gene 1.0 ST Array | *Affymetrix* - sold by *ThermoFisher* | 901168 | |
| Chemical compound, drug | 4-Aminobenzoic acid | *Sigma-Aldrich* | A9878 | |
| Chemical compound, drug | Chloroquine diphosphate salt | *Sigma-Aldrich* | C6628 | Dissolve in water, dosage: 100 mg/kg by oral gavage |
| Chemical compound, drug | Lipopolysaccharide from *Escherichia coli* 0111:B4 | *Sigma-Aldrich* | L4391 | |

*Continued on next page*

*Continued*

| Reagent type (species) or resource | Designation | Source or reference | Identifiers | Additional information |
|---|---|---|---|---|
| Software, algorithm | bowtie2 v2.2.7 | (*Langmead and Salzberg, 2012*) http://bowtie-bio.sourceforge.net/bowtie2/index.shtml | RRID:SCR_016368 | |
| Software, algorithm | DESeq2 | (*Love et al., 2014*) https://bioconductor.org/packages/release/bioc/html/DESeq2.html | RRID:SCR_015687 | |
| Software, algorithm | Cytoscape v3.8.0 | (*Shannon et al., 2003*) https://cytoscape.org/ | RRID:SCR_003032 | |
| Software, algorithm | clueGO v2.5.4 | (*Bindea et al., 2009*; *Mlecnik et al., 2014*) http://apps.cytoscape.org/apps/cluego | RRID:SCR_005748 | |
| Software, algorithm | HOMER v4.10 | (*Heinz et al., 2010*) http://homer.ucsd.edu/ | RRID:SCR_010881 | |
| Software, algorithm | Integrative genomics viewer (IGV) | (*Thorvaldsdóttir et al., 2013*) http://www.broadinstitute.org/igv/ | RRID:SCR_011793 | |

## Mice

All animal experiments were conducted in accordance with UK Home Office regulations (Animals Scientific Procedures Act 1986; project licence number 70/8546) and approved by veterinarian services at the University of Edinburgh. C57Bl/6J mice, originally obtained from the *The Jackson Laboratory*, were bred and housed in individually ventilated cages under specific pathogen-free conditions. Mice had access to water and rat and mouse no. three breeding diet (*Special Diets Services*) at all times. Experimental procedures were initiated when mice were 8–10 weeks of age, following acclimatisation to a reversed 12 hr dark/light cycle (lights OFF at 07:00 GMT and lights ON at 19:00 GMT). Mice were culled either by cervical dislocation or by pentobarbital overdose followed by exsanguination.

## Mosquito transmission of malaria parasites

We transmitted two serially blood passaged *Plasmodium chabaudi chabaudi* clones (*P. chabaudi* AS (clone 28AS11) and *P. chabaudi* AJ (clone 96AJ15)), which were originally obtained from the University of Edinburgh (http://www.malariaresearch.eu/). *Anopheles stephensi* mosquitoes (strain SD500) were reared in-house and infected with *P. chabaudi* according to our previously published protocol (*Spence et al., 2012*). In brief, donor mice were inoculated with serially blood passaged *P. chabaudi* by intraperitoneal injection of infected red cells. Gametocytes were quantified on day 14 of infection and mice with >0.1% gametocytaemia were anaesthetised and exposed to female mosquitoes at 11:00 GMT. To ensure optimal parasite development, mosquitoes were kept at 26.0°C (±0.5°C) in an ultrasonic humidity cabinet and provided with 8% Fructose and 0.05% 4-Aminobenzoic acid (*Sigma-Aldrich*) feeding solution from this point forward. Successful oocyst development was verified in mosquito midguts 8 days later and sporozoites were isolated from mosquito salivary glands on day 15 post-feed. Salivary glands were dissected under a stereomicroscope and transferred to a glass mortar; to maintain sporozoite viability, salivary glands were kept on ice in RPMI supplemented with 0.2% Glucose, 0.2% Sodium bicarbonate (*Sigma-Aldrich*), 2 mM L-Glutamine (*Gibco*), and 10% fetal bovine serum (*Gibco*, FBS Performance Plus, heat inactivated and filtered 0.22 μm) for a maximum of 2 hr. Sporozoites were released from salivary glands by gentle homogenisation and washed three times before enumeration. To initiate infection in experimental mice, 200 *P. chabaudi* sporozoites were intravenously injected into the tail vein.

## Monitoring the course and outcome of infection

Following sporozoite injection, *P. chabaudi* develops in the liver for 52 hr before the release of merozoites to kick-start asexual blood-stage replication – each blood cycle takes approximately 24 hr to complete. Mice were closely monitored for the first 14 days of acute blood-stage infection with parasitaemia quantified daily using Giemsa stained thin blood films (counting at least 10,000 red cells). Sickness behaviour and core body temperature (digital rectal thermometer, *TmeElectronics*) were also recorded daily. To assess anaemia, erythrocytes in 2 µl blood (collected from the tail tip) were counted using a Z2 Coulter Particle count and size analyser (*Beckman Coulter*). We determined the healthy range in uninfected C57Bl/6J mice housed in our facility to be $8.8–10.5 \times 10^9$ erythrocytes*ml$^{-1}$. Anaemia was classified as severe when red cell loss exceeded 50%.

Chronic infection was verified after 40 days of blood-stage parasitaemia by quantitative PCR of parasite 18S ribosomal DNA. DNA was extracted from 20 µl blood (collected from the tail tip) using Quick DNA Universal Microprep Kit (*Zymo Research*), and amplified using TaqMan Universal PCR Mastermix (*ThermoFisher*) with 9 µM of forward primer (5-GCGAGAAAGTTAAAAGAATTGA-3), 9 µM of reverse primer (5-CTAGTGAGTTTCCCCGTGTT-3), and 2.5 µM of probe ([6FAM] - AAA TTAAGCCGCAAGCTCCACG - [TAM]) on a Roche Lightcycler 480 (40 cycles of amplification). A standard curve of red cells spiked with known numbers of parasites allowed accurate quantification; mice with >5 parasites*µl$^{-1}$ (limit of detection) were considered chronically infected with *P. chabaudi*, and all other mice were excluded from the study. Chronic infection was cleared using 100 mg/kg chloroquine diphosphate salt (*Sigma-Aldrich*, dissolved in water) administered by oral gavage daily for 10 days. Memory responses were assessed 30 days after the start of chloroquine treatment, and at this time-point once-infected mice were compared to uninfected age-matched controls that received the same schedule of chloroquine treatment.

## Malaria reinfection model

Mice were first infected with *P. chabaudi* AS by intravenous (iv) injection of sporozoites, and those with qPCR-confirmed chronic parasitaemia were chloroquine treated on day 40 of blood-stage infection. Thirty days after the start of chloroquine treatment mice were infected for a second time but now by iv injection of $5 \times 10^5$ mosquito-transmitted *P. chabaudi* AJ blood-stage parasites. Reinfection was initiated by this route to avoid confounding factors that may arise as a result of liver-stage immunity, which can be observed in C57Bl/6 mice after a single infection with *P. chabaudi* AS (*Nahrendorf et al., 2015*). Note that uninfected age-matched controls received the same schedule of chloroquine treatment as reinfected mice.

## Quantification of plasma proteins by ELISA

Platelet-depleted plasma was prepared from heparinised (*Wockhardt*) blood using two consecutive centrifugation steps (1000 xg for 10 min followed by 2000 xg for 15 min). Plasma was kept cold throughout and aliquots were stored at −80˚C. We used commercially available ELISA kits to quantify plasma IFNɣ (IFNɣ mouse ELISA kit, extra sensitive, *Invitrogen*), CXCL10 (IP-10 mouse ELISA kit, *Invitrogen*), and Angiopoietin-2 (mouse/rat Angiopoietin-2 quantine ELISA kit, *R&D Systems*). Absorbance was measured using a Multiskan Ascent (*MTX Lab systems*) or FluoSTAR Omega (*BMG Labtech*) plate reader.

## Tissue preparation for histology

Spleens and femurs from *P. chabaudi*-infected mice and once-infected mice (and uninfected age-matched controls) were fixed in 10% neutral buffered Formalin (*Sigma-Aldrich*) for 24 or 48 hr, respectively. Bones were then decalcified for 48 hr using 10% EDTA (pH 7.2) with gentle shaking at 55˚C. After these steps, tissues were stored in 70% Ethanol and photographed to visualise macroscopic changes resulting from malaria. The spleen and both femurs from each mouse were paraffin-embedded in a single block and 5 µm sections were prepared (cross-section for spleen and longitudinal section for bone). Sections were stained with Hematoxylin and Eosin (H&E, *ThermoFisher*) or Prussian Blue and Neutral Red (*Scientific Laboratory Supplies* and *VWR*) at the Shared University Research Facilities, University of Edinburgh. Stained slides were assessed using a Nikon A400a bright-field microscope and images were taken with a Zeiss 503 high-resolution colour camera.

Images were cropped, white balance was adjusted and the brightness/contrast standardised between samples using Adobe Photoshop CS6.

## Parasite sequestration

To quantify the accumulation of infected red cells in the microvasculature of critical organs and tissues we used the same methodology as described in our sequestration and histopathology study (*Brugat et al., 2014*). In brief, we infected mice by intraperitoneal injection of $10^5$ mosquito-transmitted *P. chabaudi* AS or AJ blood-stage parasites and measured circulating parasitaemia at the peak of schizogony (13:00 GMT) by counting at least 5000 red cells on Giemsa stained blood films. Immediately thereafter, mice were euthanised and exsanguinated – the spleen and both femurs were prepared for histology as described above ('tissue preparation for histology'). In addition, the left lobe of the liver, the left lung and kidney, the duodenum and the heart were fixed in 10% neutral buffered Formalin (24 hr) and then stored in 70% Ethanol. All organs from each mouse were paraffin-embedded in a single block and 5 µm sections were cut and stained with H&E (longitudinal section for bone and cross section for all other tissues). The percentage of infected red cells contained within the microvasculature of every organ was quantified by counting at least 1000 red cells in at least 20 blood vessels, and is displayed relative to peripheral parasitaemia. For high-resolution images of sequestered parasites in chabaudi malaria please see *Brugat et al., 2014*.

## Flow cytometry and cell sorting

Sodium heparin (*Wockhardt*) was used as anticoagulant for whole blood samples; spleens were dissociated in C-tubes using a gentleMACS Octo Dissociator (*Miltenyi Biotec*); and bone marrow was flushed from femurs using a 27½G needle/syringe loaded with IMDM. Single cell suspensions were filtered through a 70 µm cell strainer and after red cell lysis leukocytes were counted on a haemocytometer; up to $2 \times 10^6$ cells per well were placed into a 96 well V bottom plate for staining. A Zombie Aqua Fixable Viability Dye (*BioLegend*) was used to identify dead cells, after which Fc receptors were blocked using TruStain FcX (anti-mouse CD16/32, *BioLegend*). Cell surface staining was performed at room temperature (antibody panels are detailed in *Supplementary file 1*) and for ChIPseq experiments cells were subsequently fixed in PBS with 1% paraformaldehyde and 10% FBS for 10 min at room temperature (reaction was quenched with 125 mM Glycine [*Sigma-Aldrich*]). Note that across experiments the viability of leukocytes always exceeded 93.8% (no viability stain for cell sorting). To confirm the identity of patrolling monocytes we performed an intracellular stain for the transcription factor Nr4a1 (clone 12.14, *eBioscience*) using the FoxP3/Transcription Factor Buffer Staining Set (*eBioscience*).

Cells were acquired on an LSR Fortessa flow cytometer (*BD Biosciences*) or sorted using an Aria II cell sorter (*BD Biosciences,* 85 µm nozzle, sort setting 'purity'). Both cytometers used BD FACS Diva v8 software and data were subsequently analysed using FlowJo v9 – all gating strategies are shown in *Supplementary file 1*. Note that CD115 (Csf1r) was replaced with CD11c when sorting monocytes as engagement of the Csf1 receptor has been shown to induce transcriptional changes (*Jung et al., 2000*). Samples with a sort purity <95% were excluded from the study.

The absolute number of cells in each tissue was calculated from leukocyte counts of single cell suspensions. For bone marrow, we estimated that one femur contains approximately 11% of total mouse marrow (*Colvin et al., 2004*) and extrapolated accordingly. For whole blood, we recorded the volume collected during exsanguination and then extrapolated to total circulating blood volume, according to body weight. This approach allows a direct comparison of cell numbers across tissues.

## Cytospin of red pulp macrophages

Red pulp macrophages (Lineage^neg F4/80^pos B220^neg CD11b^int CD11c^int autofluorescent cells) were flow-sorted from the spleens of uninfected mice and collected into polypropylene tubes containing IMDM supplemented with 20% FBS and 8 mM L-Glutamine. Sorted cells were then spun (1000 xg for 5 min) onto glass slides using a Shandon Cytospin 3 Cytocentrifuge (*ThermoScientific*) and stained with Prussian Blue and Neutral Red (*Sigma-Aldrich*). Red pulp macrophages were visualised and photographed using a Leica DM1000 light microscope (×100 oil objective).

## In vitro stimulation of monocytes with LPS

30,000 inflammatory monocytes (Lineage$^{neg}$ Ly6G$^{neg}$ CD11b$^{pos}$ CD11c$^{neg}$ Ly6C$^{hi}$) were flow-sorted from the spleens of chronically infected mice (*P. chabaudi* AJ) or uninfected controls and collected into polypropylene tubes containing IMDM supplemented with 5% FBS and 8 mM L-Glutamine. Following a gentle spin (450 xg for 10 min, slow brake) monocytes were resuspended in 90 µl pre-warmed IMDM containing 10% FBS and 8 mM L-Glutamine, and transferred to an ultra-low attachment 96 well flat bottom cell culture plate (*Corning*). To stimulate cells 0.3 ng LPS (Lipopolysaccharide from *Escherichia coli* 0111:B4, *Sigma-Aldrich*) was added and cells were incubated for 4 hr at 37°C and 7% CO$_2$. RNA from both adherent and non-adherent cells was preserved in 1 ml TRIzol Reagent (*ThermoFisher*) for downstream steps.

## Ex vivo RNA sequencing of monocytes

10,000 inflammatory monocytes (Lineage$^{neg}$ Ly6G$^{neg}$ CD11b$^{pos}$ CD11c$^{neg}$ Ly6C$^{hi}$) were flow-sorted from the spleens of *P. chabaudi* infected mice, once-infected mice or uninfected controls and collected into 1.5 ml eppendorf tubes containing 1 ml TRIzol Reagent. Samples were inverted ten times, incubated at room temperature for 5 min and snap frozen on dry ice; all samples were stored at −80°C prior to RNA extraction.

RNA was extracted using a modified phenol-chloroform protocol (*Chomczynski and Sacchi, 2006*) with 1-Bromo-3-chloropropane and Isopropanol (*Sigma-Aldrich* and *VWR*, respectively). Total RNA was quantified and assessed for quality and integrity by Bioanalyser (RNA Pico 6000 Chip, *Agilent*) – all sequenced samples had a RIN value above 8. cDNA was generated from 2 ng total RNA using the SMART-Seq v4 Ultra Low Input RNA Kit (*Takara Bio*) and amplified using 11 cycles of PCR. Amplified cDNA was purified using Agencourt AMPure XP beads (*Beckman Coulter*), quantified on a Qubit 2.0 Fluorometer (dsDNA HS assay, *ThermoFisher*) and quality assessed by Bioanalyser (DNA HS Kit, *Agilent*). Libraries were then constructed from 150 pg of cDNA using the Nextera XT DNA Library Preparation Kit (*Illumina*) according to the manufacturer's instructions. Libraries were quantified by Qubit (dsDNA HS assay) and fragment size distribution was assessed by Bioanalyser (DNA HS Kit). Using this information, samples were combined to create equimolar library pools that were sequenced on a NextSeq 550 platform (*Illumina*) to yield 75 bp paired-end (PE) reads; the median number of PE reads per sample passing QC across all experiments was $4.79 \times 10^7$.

## RNA sequencing analysis

FastQ files were downloaded from BaseSpace (*Illumina*) and raw sequence data assessed for quality and content using FastQC (http://www.bioinformatics.babraham.ac.uk/projects/fastqc/). We aligned paired-end sequences to the Ensembl release 96 murine transcripts set with bowtie2 v2.2.7 (*Langmead and Salzberg, 2012* parameters: —very-sensitive -p 30 —no-mixed —no-discordant —no-unal) to obtain sorted, indexed bam files. Counts for each transcript were obtained using samtools idxstats (http://www.htslib.org/doc/samtools.html) and transcript counts were imported into the R/Bioconductor environment using the DESeq2 package (*Love et al., 2014*) for pairwise comparisons. Lists of differentially expressed transcripts were filtered to retain only those with an adjusted p value ($p_{adj}$) <0.01 and a fold change >1.5 using R v3.6; multiple transcripts annotated to the same gene were consolidated by keeping the transcript with the highest absolute fold change. Heatmaps and stacked circular bar charts were generated using the R ggplot2 package (*Wickham, 2016*). All RNAseq data are publicly available (GEO accession number GSE150047).

## Functional gene enrichment analysis using clueGO

Lists of differentially expressed genes were imported into clueGO v2.5.4 (*Bindea et al., 2009*; *Mlecnik et al., 2014*) – a Cytoscape plug-in (*Shannon et al., 2003*). ClueGO identified the significantly enriched GO terms (GO Biological Process and GO Molecular Function) associated with these genes and placed them into a functionally organised non-redundant gene ontology network based on the following parameters: $p_{adj}$cutoff = 0.01; correction method used = Bonferroni step down; min. GO level = 5; max. GO level = 11; min. number of genes = 3; min. percentage = 5.0; GO fusion = true; sharing group percentage = 40.0; merge redundant groups with >40.0% overlap; kappa score threshold = 0.4; and evidence codes used [All]. Each of the functional groups was assigned a unique colour and a network was then generated using an edge-weighted spring-embedded layout

based on kappa score. We found that some GO terms were shared between multiple groups and so we manually merged these functionally connected groups to form supergroups, which we named according to the leading GO term (lowest $p_{adj}$ with min. GO level 5).

## Chromatin immunoprecipitation for sequencing (ChIPseq)

50,000 fixed inflammatory monocytes (Lineage[neg] Ly6G[neg] CD135[neg] cKit[neg] CD11b[pos] CD11c[neg] Ly6C[hi]) were flow-sorted from the bone marrow of once-infected mice or uninfected controls and collected into polypropylene tubes containing IMDM supplemented with 5% FBS and 8 mM L-Gluta-mine. Sorted monocytes were pelleted by centrifugation and washed in HBSS (*Gibco*) that was supplemented with protease inhibitors (complete ULTRA Tablets Protease Inhibitor Cocktail, *Roche*) and 5 mM sodium butyrate (*Alpha Aesar*); cell pellets were stored at −80°C. Note that for each biological sample we pooled the femurs and tibias from two mice, and three tubes (each containing 50,000 cells) were collected for every sample so that we could perform chromatin-immunoprecipitation (ChIP) with three different antibodies (H3K27ac, H3K4me1, and H3K9me3).

We performed ChIP using the True MicroChIP Kit (*Diagenode*). In brief, chromatin was sheared using a Bioruptor Pico Sonicator (five cycles: 30 s ON - 30 s OFF, fragments 100–300 bp, *Diagenode*) and 10% of sheared chromatin was kept as input control whilst 90% was immunoprecipitated overnight using antibodies against H3K27ac, H3K4me1 or H3K9me3 (all ChIP-grade from *Diagenode*). Protein-A-coated magnetic beads were then added to samples and after a 6 hr incubation unbound chromatin fragments were removed by thorough washing. ChIP and input DNA were decrosslinked and purified using MicroChIP DiaPure columns (*Diagenode*).

Libraries were prepared using the MicroPlex Library Preparation Kit v2 (*Diagenode*) and amplification was monitored in real-time on a LightCycler 480 (*Roche*) to ensure the optimum number of cycles was used. Amplified libraries were quantified by Qubit (dsDNA HS assay) and fragment size distribution assessed by Bioanalyser (DNA HS Kit). After equimolar pooling of samples we purified libraries with AMPure XP beads and sequenced on a HiSeq 4000 (75 bp PE reads) or NovaSeq S1 (100 bp PE reads) (both *Illumina*). A step-by-step guide to our optimised ChIPseq protocol is available at protocols.io (dx.doi.org/10.17504/protocols.io.bja3kign).

## ChIPseq analysis

ChIPseq data quality and content were assessed using FastQC; all samples passed initial QC and were aligned to the mm10 genome using bowtie2 v2.2.7 (parameters: —very-sensitive -p 30 —no-mixed —no-discordant —no-unal). We then used the motif discovery software HOMER (v4.10 *Heinz et al., 2010*) to turn indexed bam files (generated using samtools idxstats) into tag directories of individual ChIP and input samples. Alignments were converted to bedgraph format using the HOMER script makeUCSCfile. Wig format outputs were converted to tdf files to view data in the Integrative Genomics Viewer (IGV v.2.7.2 *Thorvaldsdóttir et al., 2013*) using igvtools v2.3.93 (parameters: toTDF -z 7 f p98). In HOMER, we identified areas of the genome where ChIP read counts were significantly enriched over fragmented, non-immunoprecipitated input DNA (which indicates the presence of a histone mark) by calling peaks in ChIP relative to sample-matched input DNA using predefined parameters (H3K27ac and H3K9me3 used 'regions' and H3K4me1 used 'typical' and 'supertypical'). Default settings were used in every case with the exception of fold change over input, which was set to >3 fold for H3K9me3.

Individual samples were then combined in HOMER to create pooled ChIP tag directories. We selected only samples with a high IP efficiency (>5% for H3K27ac and H3K9me3, and >10% for H3K4me1) to generate pooled tag directories that comprised: four biological replicates for H3K27ac; 2 (once-infected) or 3 (uninfected) biological replicates for H3K4me1; and one replicate for H3K9me3. Bedgraphs of pooled ChIP tag directories were generated and converted to tdf format as above using more than $2.2 \times 10^8$ tags for H3K27ac, $1.1 \times 10^8$ tags for H3K4me1 and $7 \times 10^7$ tags for H3K9me3. Peaks were called on pooled ChIP samples relative to pool-matched input DNA using the parameters described above. We created tdf files that indicated the position of peaks across the genome with a fixed height bar and visualised the histone modification profile alongside peak location in IGV. To ask whether genes were marked or not marked we looked for the presence or absence of peaks within 10 kb (H3K27ac and H3K9me3) or 100 kb (H3K4me1) of the transcription start site.

To identify differentially modified regions (DMR) across the genome we again called peaks on pooled ChIP samples but this time instead of using non-immunoprecipitated input DNA to correct for background we called peaks in once-infected mice relative to uninfected controls (and vice versa). In this way, we identified areas of the genome where read counts were significantly enriched in one or the other experimental group. Low confidence peaks were removed by applying a peak score cut off >3 and DMR were annotated to the nearest gene using the script annotatePeaks in HOMER. We then asked how many of the 2848 tolerised/specialised genes identified by RNAseq were annotated with a differentially modified region (if a gene was annotated with more than one DMR then the region with the highest peak score was retained). All ChIPseq data (inc individual biological replicates and pooled tag directories) are publicly available (GEO accession number GSE150478).

### Ex vivo transcriptional profiling of red pulp macrophages

10,000 red pulp macrophages (Lineage$^{neg}$ F4/80$^{pos}$ B220$^{neg}$ CD11b$^{int}$ CD11c$^{int}$ autofluorescent cells) were flow-sorted from mice 100 days after self-resolving *P. chabaudi* infection or from uninfected age-matched controls. Sorted cells were collected into 1.5 ml eppendorf tubes containing 1 ml TRIzol Reagent (*ThermoFisher*) and samples were stored at −80℃ prior to RNA extraction. RNA was extracted using a modified phenol-chloroform protocol (*Chomczynski and Sacchi, 2006*) and treated with Baseline-ZERO DNase to remove genomic DNA (*Illumina*). DNase-treated RNA was then purified using the RNA Clean and Concentrator-5 Kit (*Zymo Research*) and total RNA was quantified and assessed for quality and integrity by Bioanalyser (RNA Pico 6000 Chip, *Agilent*). RNA samples were processed for gene expression analysis using the GeneChip WT Pico Kit and Mouse Gene 1.0 ST Array (*Affymetrix*) according to the manufacturer's instructions.

Microarray data were processed in R/Bioconductor making use of the oligo, pd.mta.1.0 and mta10sttranscriptcluster packages. Data quality was assessed using the arrayQualityMetrics package (*Kauffmann and Huber, 2010*); all samples passed QC and were normalised using robust multi-array average (RMA), which results in log2 expression intensities. Limma (linear models for microarray data) and eBayes packages were used for pairwise comparisons to find differentially expressed genes (DEG) between groups (AS vs uninfected, AJ vs uninfected and AJ vs AS) but yielded zero DEG in all comparisons ($p_{adj}$ <0.05). Log2 expression intensities of signature genes for monocytes, tissue resident macrophages and dendritic cells were plotted using the heatmap.2() function in R; these genelists were manually compiled from published studies that set out to identify the gene expression profiles that underpin identity in myeloid cells (*Gautier et al., 2012*; *Haldar et al., 2014*; *Miller et al., 2012*; *Okabe and Medzhitov, 2014*). Microarray data are publicly available (GEO accession number GSE149894).

### Data access

All RNAseq, ChIPseq, and microarray data have been deposited in NCBI's Gene Expression Omnibus (*Edgar et al., 2002*) and are accessible through GEO SuperSeries accession number GSE150479.

### Acknowledgements

We thank Ronnie Mooney for the production of *Anopheles stephensi* mosquitoes and his assistance with in vivo experiments. Flow cytometry data were generated within the cell sorting facility in Ashworth laboratories (University of Edinburgh), which is supported by funding from the Wellcome Trust. RNA sequencing libraries were prepared and sequenced by the Edinburgh Clinical Research Facility at the University of Edinburgh, which receives financial support from NHS Research Scotland (NRS). ChIPseq libraries were sequenced by Edinburgh Genomics, which is supported through core grants from NERC (R8/H10/56), MRC UK (MR/K001744/1) and BBSRC (BB/J004243/1). We thank the Shared University Research Facilities at Little France for histology and Hologic Ltd. for microarray services. This project was supported by the Wellcome Trust-University of Edinburgh Institutional Strategic Support Fund, and PJS is the recipient of a Sir Henry Dale Fellowship jointly funded by the Wellcome Trust and the Royal Society (grant no. 107668/Z/15/Z). We are grateful to Steve Jenkins, Eleanor Riley and Joanne Thompson for their thoughtful comments on our manuscript.

## Additional information

### Funding

| Funder | Grant reference number | Author |
|---|---|---|
| Wellcome Trust | Sir Henry Dale Fellowship (107668/Z/15/Z) | Philip J Spence |
| Royal Society | Sir Henry Dale Fellowship (107668/Z/15/Z) | Philip J Spence |
| Wellcome Trust | University of Edinburgh Institutional Strategic Support Fund | Philip J Spence |

The funders had no role in study design, data collection and interpretation, or the decision to submit the work for publication.

### Author contributions

Wiebke Nahrendorf, Conceptualization, Formal analysis, Investigation, Visualization, Methodology, Writing - original draft, Writing - review and editing,; Alasdair Ivens, Data curation, Software, Formal analysis, Writing - review and editing; Philip J Spence, Conceptualization, Formal analysis, Supervision, Funding acquisition, Investigation, Methodology, Writing - original draft, Writing - review and editing

### Author ORCIDs

Wiebke Nahrendorf (iD) https://orcid.org/0000-0002-4503-8761
Philip J Spence (iD) https://orcid.org/0000-0002-5506-2773

### Ethics

Animal experimentation: All animal experiments were conducted in strict accordance with UK Home Office regulations (Animals Scientific Procedures Act 1986; project licence number 70/8546) and approved by veterinarian services at the University of Edinburgh. Ethical approval for this study was provided by the Animal Welfare and Ethical Review Body (AWERB) and the School of Biological Sciences Ethics Committee at the University of Edinburgh.

### Decision letter and Author response

Decision letter https://doi.org/10.7554/eLife.63838.sa1
Author response https://doi.org/10.7554/eLife.63838.sa2

## Additional files

### Supplementary files

• Supplementary file 1. Gating strategies for flow cytometry and cell sorting. Flow cytometry was performed using the listed antibodies and panels. We gated myeloid cells and their progenitors in bone marrow, blood, and spleen using FlowJo v9; flow profiles of uninfected mice are displayed alongside the acute phase of a first malaria episode (*P. chabaudi* AJ). In every case, gating was performed identically between uninfected and infected mice with one exception (marked with an asterisk); to identify myeloid and erythroid progenitors in the bone marrow of infected mice we had to adjust our first gate (on lineage negative live singlets) due to the well-known upregulation of Sca-1 during acute infection (*Belyaev et al., 2010*). Note that CD115 (Csf1r) was replaced with CD11c when sorting monocytes as engagement of the Csf1 receptor has been shown to induce transcriptional changes (*Jung et al., 2000*).

• Supplementary file 2. Quantitative changes in the histone modification profiles of once-infected mice. Bone marrow monocytes were flow-sorted from once-infected mice (AJ, memory, 70 days p.i.) and uninfected controls for chromatin immunoprecipitation (ChIP)seq; differences in their histone modification profiles were then quantified by calling differentially modified regions (DMR, annotated

to the nearest gene). Shown is a list of all tolerised/specialised genes annotated with a DMR, ordered by peak score.

• Transparent reporting form

## Data availability

All RNAseq, ChIPseq and microarray data have been deposited in NCBI's Gene Expression Omnibus and are accessible through GEO SuperSeries accession number GSE150479.

The following datasets were generated:

| Author(s) | Year | Dataset title | Dataset URL | Database and Identifier |
|---|---|---|---|---|
| Nahrendorf W, Ivens A, Spence PJ | 2020 | A single malaria episode induces mechanisms that minimise inflammation and promote tolerance in spleen inflammatory monocytes | https://www.ncbi.nlm. nih.gov/geo/query/acc. cgi?acc=GSE150047 | NCBI Gene Expression Omnibus, GSE150047 |
| Nahrendorf W, Ivens A, Spence PJ | 2020 | Bone marrow monocytes from once-malaria infected mice have no epigenetic memory of the infection | https://www.ncbi.nlm. nih.gov/geo/query/acc. cgi?acc=GSE150478 | NCBI Gene Expression Omnibus, GSE150478 |
| Nahrendorf W, Ivens A, Spence PJ | 2020 | Tissue printing: splenic red pulp macrophages of once-malaria infected mice are transcriptionally identical to prenatally seeded red pulp macrophages from uninfected mice | https://www.ncbi.nlm. nih.gov/geo/query/acc. cgi?acc=GSE149894 | NCBI Gene Expression Omnibus, GSE149894 |
| Nahrendorf W, Ivens A, Spence PJ | 2020 | Inducible mechanisms of disease tolerance provide an alternative strategy of acquired immunity to malaria. | https://www.ncbi.nlm. nih.gov/geo/query/acc. cgi?acc=GSE150479 | NCBI Gene Expression Omnibus, GSE150479 |

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
