## [Decision Letter]

**Acceptance summary:**

This study investigated the phenomenon of long-term tolerance induced by malaria infection, a relevant and important aspect of pathophysiology of malaria infection. This study provides novel important data to understand this process, proposing that innate immune memory contributes to the induction of long-term tolerance after a malaria infection, subsequently improving the survival of the host. The subject is very relevant, as post-malaria tolerance is one of the most important mechanisms improving outcome of this severe disease.

**Decision letter after peer review:**

Thank you for submitting your article "Inducible mechanisms of disease tolerance provide an alternative strategy of acquired immunity to malaria" for consideration by *eLife*. Your article has been reviewed by Satyajit Rath as the Senior Editor, a Reviewing Editor, and three reviewers. The reviewers have opted to remain anonymous.

The reviewers have discussed the reviews with one another and the Reviewing Editor has drafted this decision to help you prepare a revised submission.

Summary:

All reviewers agreed on the interesting and relevant aspect of this study aiming to understand why repeated exposure to malaria results to long-term tolerance of the parasite. The authors indeed propose here that innate immune memory contributes to the induction of long-term tolerance after a malaria infection, subsequently improving the survival of the host.

Essential revisions:

However, some aspects of the study need to be improved to strengthen the conclusions made here. Key comments/experiments are listed below:

To clarify by which markers the authors define their subsets of interest, such as monocytes or “specialized iron-recycling macrophages”. In Figure 1, What are inflammatory monocytes, how are they characterized? What are myeloid progenitors? Are these GMPs? cMOPS? No exact definition or gating strategy is provided. Have the authors excluded that the differences in myeloid cell progenitors are on the subset level and or on the level of functional surface molecules, such as CD131, as suggested in Mitroulis et al., 2018? These changes might not be evident in a general myeloid progenitor quantification. In this regard, the detailed phenotype of myeloid BM progenitors during tolerance induction should be provided.

The authors need to show that the parasite burden is equivalent between the primary and secondary infections. This is crucial to the interpretation of the findings. In Figure 3B, the parasitemia is only shown for day 6, likely the start of the primary parasitemia curve for this infection. The authors need to display the full parasitemia curves for both the first and second infections, to demonstrate that the parasite burden is equal (or very similar) over the full timecourse in the first and second infections.

The lack of epigenetic reprogramming in circulating monocytes is remarkable and it is indeed a potential argument that induction of long-term immune tolerance is taking place in the spleen, as hypothesize by the authors. But all reviewers think that the authors should present some basic data demonstrating long-term epigenetic/transcriptional reprogramming of spleen macrophages. Without such data, the title of Figure 6 is misleading.

All reviewers recommended to perform crucial experiments on splenectomized mice to support that spleen is crucial for the induction and maintenance of the tolerance. In addition, crucial experiments should also be done in RAG mice to at least control for adaptive immune-mediated control of the parasites, which may be significant. Indeed, given that AJ infection has a higher parasite burden in the primary infection (Figure 1B), but the parasitemias are described as equal between the primary and secondary infections in the model used, it is likely that there is substantial adaptive immune control of the parasites in the second infection. This is a confounding factor that is not mentioned at all in the manuscript but is likely to have a major impact on the way that the parasites are sensed in the second infection. Do opsonised merozoites inhibit inflammation in the myeloid population, when compared to non-opsonised merozoites or infected red blood cells? This may provide an alternative explanation for the phenomenon observed here. What about the role of regulatory populations? Would similar results be observed if the authors performed these experiments in RAG or Nude mice following drug cure of the first infection? Would transfer of monocytes or macrophages (if feasible) from the spleen of cured mice confer a level of tolerance to new hosts?

---

## [Author Response]

Essential revisions:However, some aspects of the study need to be improved to strengthen the conclusions made here. Key comments/experiments are listed below:To clarify by which markers the authors define their subsets of interest, such as monocytes or “specialized iron-recycling macrophages”. In Figure 1, What are inflammatory monocytes, how are they characterized? What are myeloid progenitors? Are these GMPs? cMOPS? No exact definition or gating strategy is provided. Have the authors excluded that the differences in myeloid cell progenitors are on the subset level and or on the level of functional surface molecules, such as CD131, as suggested in Mitroulis et al., 2018? These changes might not be evident in a general myeloid progenitor quantification. In this regard, the detailed phenotype of myeloid BM progenitors during tolerance induction should be provided.

We agree that a clear definition and gating strategy for each myeloid and progenitor cell population is essential and had included this in our original submission. We apologise that we did not effectively draw attention to this resource and have modified our revised manuscript to make it much easier for the reader to find this information. Details of all antibodies, staining panels and gating strategies in both uninfected and AJ-infected mice are provided in Supplementary file 1. We now explicitly refer to this in the Results (“see Supplementary file 1 for gating strategies”), all figure legends that include flow data and the Materials and methods (“antibody panels are detailed in Supplementary file 1” and “gating strategies are shown in Supplementary file 1”). To identify myeloid cells by flow cytometry we used the following gating strategies:

granulocyte monocyte progenitors (GMP)

(Lineage^neg^ Sca1^neg^ c-Kit^pos^ CD34^pos^ CD16/32 (FcɣRIII/II)^pos^ CD27^pos^ CD71^int^)

megakaryocyte erythroid progenitors (MEP)

(Lineage^neg^ Sca1^neg^ c-Kit^pos^ CD34^neg^ CD16/32 (FcɣRIII/II)^neg^ CD27^neg^ CD71^pos^)

inflammatory monocytes bone marrow *

(Lineage^neg^ Ly6G^neg^ CD115^pos^ CD135^neg^ cKit^neg^ CD11b^pos^ Ly6C^hi^)

* note that this gating strategy specifically excludes immature monocyte precursors like the common monocyte progenitor (cMoP: Lineage^neg^ Ly6G^neg^ CD115^pos^ CD135^neg^ cKit^pos^ CD11b^int^ Ly6C^hi^) and the macrophage dendritic cell progenitor (MDP: Lineage^neg^ Ly6G^neg^ CD115^pos^ CD135^neg^ cKit^pos^ CD11b^lo^ Ly6C^lo^)

inflammatory monocytes spleen

(Lineage^neg^ Ly6G^neg^ CD115^pos^ CD11b^pos^ Ly6C^hi^)

patrolling monocytes

(Lineage^neg^ CD115^pos^ CD11b^pos^ Ly6C^neg^ Nr4a1^pos^ (most are also CX3CR1^pos^)

red pulp macrophages

(Lineage^neg^ F4/80^pos^ B220^neg^ CD11b^int^ CD11c^int^ autofluorescent cells)

bone marrow iron recycling macrophages

(Lineage^neg^ CD115^pos^ CD11b^int^ CD169^pos^ CD11c^int^ F4/80^pos^)

We devised these gating strategies according to the following key publications: Hey, 2015; Hettinger, 2013; Hanna, 2011; Hashimoto, 2013; and Paul, 2015.

In the Materials and methods, we specify the phenotype of every cell population flow-sorted for RNA sequencing, ChIPseq, microarray, cytospin and in vitro stimulation and clearly state that CD115 was not used when sorting monocytes: “Note that CD115 (Csf1r) was replaced with CD11c when sorting monocytes as engagement of the Csf1 receptor has been shown to induce transcriptional changes (Jung et al., 2000).”

We have also revised Figure 1 (and the accompanying legend) to clearly state that the myeloid progenitors that we quantify are GMP – their absolute number and proportion (relative to MEP) both increase during acute infection in naive mice. In a previous publication, Mitroulis et al., (2018) showed that myeloid cell production can also be increased in the bone marrow by injecting mice with β glucan. In this setting, myeloid expansion is driven by an increased frequency of CD131-expressing multipotent progenitors and haematopoietic stem cells. Unfortunately, we did not stain for CD131 and so are unable to comment as to whether emergency myelopoiesis in malaria may also be underpinned by changes in uncommitted progenitors upstream of GMPs.

The authors need to show that the parasite burden is equivalent between the primary and secondary infections. This is crucial to the interpretation of the findings. In Figure 3B, the parasitemia is only shown for day 6, likely the start of the primary parasitemia curve for this infection. The authors need to display the full parasitemia curves for both the first and second infections, to demonstrate that the parasite burden is equal (or very similar) over the full timecourse in the first and second infections.

We completely agree; matching parasite densities between first and second infection is absolutely essential to identify acquired mechanisms of disease tolerance. In our original submission the data in Figure 3B represented the parasitaemia of those mice sacrificed for RNA sequencing of spleen monocytes on day 7 of first infection and reinfection (Figure 2 and Figure 4, respectively). We chose to display the data in that way to emphasise that the specialised functions of monocytes during reinfection could not be explained by a difference in pathogen load. However, we acknowledge that we did not communicate our reasoning in the manuscript and agree that showing a full course of infection would be more informative and transparent. We have therefore revised Figure 3B to show all data across the acute phase of first and second infection – importantly, this contains every time-point at which we performed analysis of the host response (including flow cytometry, histology, plasma analyte profiling and RNA sequencing). Furthermore, we now show the data as parasite density – calculated using the percentage parasitaemia and red cell counts for each mouse at each time-point – to give a more accurate measure of pathogen load. As can be seen in the revised figure, parasite burden is very similar throughout the acute phase of first and second infection. Moreover, there is no statistically significant difference at any time-point:

“No statistically significant difference was detected at any timepoint (p_adj_ < 0.05, Mann-Whitney test corrected for multiple comparisons using Holm-Šidák method).”

Note that because we sacrificed mice for analysis along the way it would be misleading to represent the data as a continuous curve and we therefore show individual data points. Also note that Figure 3B shows chronically infected mice in second infection have a similar parasite burden to chronically infected mice in first infection (day 40 post-infection). We hope that by showing these data in full we are able to communicate the lengths to which we went to ensure pathogen load could not be a confounding factor in our study.

The lack of epigenetic reprogramming in circulating monocytes is remarkable and it is indeed a potential argument that induction of long-term immune tolerance is taking place in the spleen, as hypothesize by the authors. But all reviewers think that the authors should present some basic data demonstrating long-term epigenetic/transcriptional reprogramming of spleen macrophages. Without such data, the title of Figure 6 is misleading.

The absence of epigenetic reprogramming of bone marrow monocytes is remarkable (and surprising) and we believe this result will be extremely informative in the emerging field of innate memory. To emphasise the importance of this finding (and to more accurately state the conclusion supported by these data) we have changed the title of Figure 6 and the corresponding subheading in the Results section to: “Malaria does not induce epigenetic reprogramming of bone marrow monocytes”. This removes any reference to imprinting of monocyte function in the spleen, which could be misleading.

We now put forward the possibility that monocyte fate may be imprinted within the remodelled spleen in a new Results section. This section presents new data (Figure 7) showing that monocytes are transcriptionally reprogrammed in the spleen of once-infected mice. Importantly, this is exactly the same time-point at which we performed ChIPseq on bone marrow monocytes (Figure 6) and all of our reinfection studies. In brief, we found a persisting transcriptional signature of infection in spleen monocytes 30-days after drug cure. Remarkably, most of these genes were not differentially expressed during acute or chronic infection; instead, this transcriptional signature was unique to the memory phase and was further enhanced upon reinfection. GO term analysis then confirmed that many of the functional gene enrichment pathways unique to second infection were already present in spleen monocytes prior to reinfection. We believe these data provide strong support for the argument that the functional specialisation of monocytes in tolerised hosts takes place within the remodelled spleen. However, we acknowledge that these data do not provide definitive evidence and this is reflected in our choice of language throughout the manuscript, for example: Abstract “this alternative fate […] appears to be imprinted within the remodelled spleen”; Results “malaria may remodel the spleen to imprint tolerance”; Figure 7C “we therefore propose that the remodelled spleen imprints monocytes with tissue protective functions”; and Discussion “functional specialisation in tolerised hosts […] seems to be imprinted within the remodelled spleen”. We also explicitly state in the discussion that further experiments are required to provide definitive evidence that monocyte fate is imprinted within the spleen, as discussed in detail below.

All reviewers recommended to perform crucial experiments on splenectomized mice to support that spleen is crucial for the induction and maintenance of the tolerance. In addition, crucial experiments should also be done in RAG mice to at least control for adaptive immune-mediated control of the parasites, which may be significant. Indeed, given that AJ infection has a higher parasite burden in the primary infection (Figure 1B), but the parasitemias are described as equal between the primary and secondary infections in the model used, it is likely that there is substantial adaptive immune control of the parasites in the second infection. This is a confounding factor that is not mentioned at all in the manuscript but is likely to have a major impact on the way that the parasites are sensed in the second infection. Do opsonised merozoites inhibit inflammation in the myeloid population, when compared to non-opsonised merozoites or infected red blood cells? This may provide an alternative explanation for the phenomenon observed here. What about the role of regulatory populations? Would similar results be observed if the authors performed these experiments in RAG or Nude mice following drug cure of the first infection? Would transfer of monocytes or macrophages (if feasible) from the spleen of cured mice confer a level of tolerance to new hosts?

Two important questions are raised here: can we conclusively state that monocytes are functionally reprogrammed in the remodelled spleen? And what role does adaptive immunity play in disease tolerance?

We agree that effective mechanisms of adaptive immunity are operating in our reinfection model, reducing the peak of *P. chabaudi* AJ parasitaemia from 25% in first infection to 0.5% in second infection. This reduction in parasitaemia is an example of host resistance (improved clearance of parasites) and is presumably mediated by acquired mechanisms of humoral and/or T cell immunity. But as we show using the less virulent genotype AS 0.5% parasitaemia is still sufficient to promote systemic inflammation, tissue damage and the differentiation of inflammatory macrophages in naive hosts. The question should therefore be slightly re-framed – can adaptive immunity also contribute to acquired mechanisms of disease tolerance?

We have included a new paragraph in the Discussion that explores this question in detail. In brief, we think it unlikely that opsonising antibodies can explain the specialised functions of monocytes in tolerised hosts as phagocytosis of immune complexes or opsonised infected red cells leads to increased production of IL-1β, IL-6 and TNF in human monocytes and macrophages (Osier, 2014 and Zhou, 2012). An alternative explanation may be that memory T cells modify the availability of key signals that can regulate monocyte fate (e.g. IFNɣ). Changing the availability of one cytokine may not be enough to drive the diverse host adaptations that we observe in second infection but more extensive changes in T cell function could make a significant contribution to disease tolerance. In the revised manuscript, we clearly state that adaptive mechanisms of immunity are operating in our reinfection model and provide partial host resistance. Furthermore, we propose that changes in T cell cytokine production may complement structural remodelling to create a tissue environment that can redirect monocyte fate; importantly, we point out that this hypothesis requires additional experimentation.

As suggested by the reviewers, one way to investigate the role of adaptive immunity in disease tolerance is to use RAG KO or nude mice. Unfortunately, we do not feel that these experiments can satisfactorily answer the questions raised. Infection of RAG KO mice leads to hyperparasitaemia and death within 8-10 days even when using the avirulent genotype AS (Spence, 2013). Drug cure would therefore have to be initiated extremely early and this model would not recapitulate the chronic recrudescing parasitaemia that is observed in human malaria. It is not clear whether disease tolerance could be established in this setting. An alternative to infecting immunodeficient mice would be to use antibodies to deplete T cells from wild-type mice prior to reinfection but this would lead to hyperparasitaemia upon rechallenge. We would therefore expect disease severity to increase in second infection but it would be impossible to know whether this results from a loss of tolerance or resistance mechanisms.

In much the same way, splenectomy would not be able to directly address whether monocyte fate is imprinted within the remodelled spleen. In this scenario, we would remove the only organ that can mechanically trap and filter infected red cells, and mice would therefore have an increased pathogen load in second infection (Safeukui, 2008). This has been clearly shown in *P. chabaudi* (Grun, 1985) and in semi-immune adults living in endemic areas (Bach, 2005). As pointed out by the reviewers, parasite burden needs to be equivalent between first and second infection to interrogate mechanisms of disease tolerance (“this is crucial to the interpretation of the findings”). We should also point out that splenectomy can alter the expression of parasite variant surface antigens leading to major changes in sequestration and histopathology (Del Portillo, 2012). Taking all of this into account, we would predict that tolerance would be broken by splenectomy but this would not conclusively show that monocyte fate needs to be imprinted within the remodelled spleen – our results would be confounded by hyperparasitaemia and changing sequestration patterns.

Instead, we propose that transplanting spleens from once-infected to naive mice (and then infecting recipient mice with *P. chabaudi* AS) may be a better route through which to ask whether monocyte function is imprinted within the remodelled spleen. Nevertheless, we must caution that in this setting none of the other mechanisms of disease tolerance will be in place and it is possible that by triggering emergency myelopoiesis, systemic inflammation and endothelium activation recipient mice may override local mechanisms of tolerance that operate in the spleen. The same caveat applies to the adoptive transfer of spleen monocytes. Furthermore, these experiments would only reveal whether monocyte fate is imprinted within the spleen and would not provide any additional information that may help elucidate the relative contribution of structural changes versus T cell regulation. This could be teased apart in a transplantion model by giving drug-treated donor mice T cell depleting antibodies prior to removal of the spleen. Here, recipient mice would receive a remodelled spleen but no malaria-specific memory T cells. This is one reason why we favour spleen transplants over adoptive transfers and this is discussed at length in the revised manuscript.

We have given all of these experimental approaches a lot of thought over the last few years but still feel that there is no simple experiment that can give a straightforward answer. The data presented in this study clearly show that disease tolerance is underpinned by diverse host adaptations that work together at whole-organism scale to provide long-lived clinical immunity. Changes in the myeloid compartment are likely to be accompanied by changes to the adaptive immune response and further complemented by metabolic adaptations (Discussion). The complex interplay between all of these adaptations will make it very difficult to dissect each mechanism in isolation. As such, we believe that a new dedicated programme of work will be required to take these next steps. Moreover, we believe that the absence of definitive evidence for imprinting in the spleen does not detract from the remarkable finding that there is no epigenetic reprogramming in the bone marrow.